# IMPROVING ADVERSARIAL ROBUSTNESS BY PUTTING MORE REGULARIZATIONS ON LESS ROBUST SAMPLES

## ABSTRACT

Adversarial training, which is to enhance robustness against adversarial attacks, has received much attention because it is easy to generate human-imperceptible perturbations of data to deceive a given deep neural network. In this paper, we propose a new adversarial training algorithm that is theoretically well motivated and empirically superior to other existing algorithms. A novel feature of the proposed algorithm is to apply more regularization to data vulnerable to adversarial attacks than other existing regularization algorithms do. Theoretically, we show that our algorithm can be understood as an algorithm of minimizing a newly derived upper bound of the robust risk. Numerical experiments illustrate that our proposed algorithm improves the generalization (accuracy on examples) and robustness (accuracy on adversarial attacks) simultaneously to achieve the state-of-the-art performance.

## 1 INTRODUCTION

It is easy to generate human-imperceptible perturbations that put prediction of a deep neural network (DNN) out. Such perturbed samples are called *adversarial examples* (Szegedy et al., 2014) and algorithms for generating adversarial examples are called *adversarial attacks*. It is well known that adversarial attacks can greatly reduce the accuracy of DNNs, for example from about 96% accuracy on clean data to almost zero accuracy on adversarial examples (Madry et al., 2018). This vulnerability of DNNs can cause serious security problems when DNNs are applied to security critical applications (Kurakin et al., 2017; Jiang et al., 2019) such as medicine (Ma et al., 2020; Finlayson et al., 2019) and autonomous driving (Kurakin et al., 2017; Deng et al., 2020; Morgulis et al., 2019; Li et al., 2020).

Adversarial training, which is to enhance robustness against adversarial attacks, has received much attention. Various adversarial training algorithms can be categorized into two types. The first one is to learn prediction models by minimizing the robust risk - the risk for adversarial examples. PGD-AT (Madry et al., 2018) is the first of its kinds and various modifications including Zhang et al. (2020); Ding et al. (2020); Zhang et al. (2021) have been proposed since then.

The second type of adversarial training algorithms is to minimize the regularized risk which is the sum of the empirical risk for clean examples and a regularized term related to adversarial robustness. TRADES (Zhang et al., 2019) decomposes the robust risk into the sum of the natural and boundary risks, where the first one is the risk for clean examples and the second one is the remaining part, and replaces them to their upper bounds to have the regularized risk. HAT (Rade & Moosavi-Dezfolli, 2022) modifies the regularization term of TRADES by adding an additional regularization term based on helper samples.

The aim of this paper is to develop a new adversarial training algorithm for DNNs, which is theoretically well motivated and empirically superior to other existing competitors. Our algorithm modifies the regularization term of TRADES (Zhang et al., 2019) to put more regularization on less robust samples. This new regularization term is motivated by an upper bound of the boundary risk.

Our proposed regularized term is similar to that used in MART (Wang et al., 2020). The two key differences are that (1) the objective function of MART consists of the sum of the robust risk and regularization term while ours consists of the sum of the natural risk and regularization term and (2) our algorithm regularizes less robust samples more but MART regularizes less accurate samples more. Note that our algorithm is theoretically well motivated from an upper bound of the robust risk

but no such theoretical explanation of MART is available. In numerical studies, we demonstrate that our algorithm outperforms MART as well as TRADES with large margins.

## 1.1 Our Contributions

We propose a new adversarial training algorithm. Novel features of our algorithm compared to other existing adversarial training algorithms are that it is theoretically well motivated and empirically superior. Our contributions can be summarized as follows:

- We derive an upper bound of the robust risk for multi-classification problems.
- As a surrogate version of this upper bound, we propose a new regularized risk.
- We develop an adversarial training algorithm that learns a robust prediction model by minimizing the proposed regularized risk.
- By analyzing benchmark data sets, we show that our proposed algorithm is superior to other competitors in view of the generalization (accuracy on clean examples) and robustness (accuracy on adversarial examples) simultaneously to achieve the state-of-the-art performance.
- We illustrate that our algorithm is helpful to improve the fairness of the prediction model in the sense that the error rates of each class become more similar compared to TRADES.

## 2 Preliminaries

### 2.1 Robust Population Risk

Let $\mathcal{X} \subset \mathbb{R}^d$ be the input space, $\mathcal{Y} = \{1, \cdots, C\}$ be the set of output labels and $f_{\boldsymbol{\theta}} : \mathcal{X} \to \mathbb{R}^C$ be the score function parameterized by the neural network parameters $\boldsymbol{\theta}$ (the vector of weights and biases) such that $\mathbf{p}_{\boldsymbol{\theta}}(\cdot|\boldsymbol{x}) = \mathrm{softmax}(f_{\boldsymbol{\theta}}(\boldsymbol{x}))$ is the vector of the conditional class probabilities. Let $F_{\boldsymbol{\theta}}(\boldsymbol{x}) = \arg\max_c [f_{\boldsymbol{\theta}}(\boldsymbol{x})]_c$, $\mathcal{B}_p(\boldsymbol{x}, \varepsilon) = \{\boldsymbol{x}' \in \mathcal{X} : \|\boldsymbol{x} - \boldsymbol{x}'\|_p \leq \varepsilon\}$ and $\mathbb{1}(\cdot)$ be the indicator function. Let capital letters $\mathbf{X}, \mathbf{Y}$ denote random variables or vectors and small letters $\boldsymbol{x}, y$ denote their realizations.

The robust population risk used in the adversarial training is defined as

$$\mathcal{R}_{\mathrm{rob}}(\theta) := \mathbb{E}_{(\mathbf{X}, \mathbf{Y})} \max_{\mathbf{X}' \in \mathcal{B}_p(\mathbf{X}, \varepsilon)} \mathbb{1}\left\{F_{\boldsymbol{\theta}}(\mathbf{X}') \neq \mathbf{Y}\right\}, \tag{1}$$

where $\mathbf{X}$ and $\mathbf{Y}$ are a random vector in $\mathcal{X}$ and a random variable in $\mathcal{Y}$, respectively. Most adversarial training algorithms learn $\boldsymbol{\theta}$ by minimizing an empirical version of the above robust population risk. In turn, most empirical versions of (1) require to generate an *adversarial example* which is a surrogate version of

$$\boldsymbol{x}^{\mathrm{adv}} := \arg\max_{\boldsymbol{x}' \in \mathcal{B}_p(\boldsymbol{x}, \varepsilon)} \mathbb{1}\left\{F_{\theta}(\boldsymbol{x}') \neq y\right\}.$$

Any method of generating an adversarial example is called an *adversarial attack*.

### 2.2 Algorithms for Generating Adversarial Examples

Existing adversarial attacks can be categorized into either the white-box attack (Goodfellow et al., 2015; Madry et al., 2018; Carlini & Wagner, 2017; Croce & Hein, 2020a) or the black-box attack (Papernot et al., 2016; 2017; Chen et al., 2017; Ilyas et al., 2018; Papernot et al., 2018). For the white-box attack, the model structure and parameters are known to adversaries who use this information for generating adversarial examples, while outputs for given inputs are only available to adversaries for the black-box attack.

The most popular method for the white-box attack is PGD (Projected Gradient Descent) (Madry et al., 2018). Let $\eta(\boldsymbol{x}'|\theta, \boldsymbol{x}, y)$ be a surrogate loss of $\mathbb{1}\{F_{\theta}(\boldsymbol{x}') \neq y\}$ for given $\boldsymbol{\theta}, \boldsymbol{x}, y$. PGD finds the adversarial example by applying the gradient ascent algorithm to $\eta$ to update $\boldsymbol{x}'_{\eta}$ and projecting it to $\mathcal{B}_p(\boldsymbol{x}, \varepsilon)$. That is, the update rule of PGD is

$$\boldsymbol{x}^{(t+1)} = \boldsymbol{\Pi}_{\mathcal{B}_p(\boldsymbol{x}, \varepsilon)} \left( \boldsymbol{x}^{(t)} + \nu \, \mathrm{sgn}\left( \nabla_{\boldsymbol{x}^{(t)}} \eta(\boldsymbol{x}^{(t)}|\boldsymbol{\theta}, \boldsymbol{x}, y) \right) \right), \tag{2}$$

where $\nu > 0$ is the step size, $\mathbf{\Pi}_{\mathcal{B}_p(\boldsymbol{x},\varepsilon)}(\cdot)$ is the projection operator to $\mathcal{B}_p(\boldsymbol{x},\varepsilon)$ and $\boldsymbol{x}^{(0)} = \boldsymbol{x}$. We define $\boldsymbol{x}^{\mathrm{pgd}}$ as $\boldsymbol{x}^{\mathrm{pgd}} := \lim_{t \to \infty} \boldsymbol{x}^{(t)}$. For the surrogate loss $\eta$, the cross entropy (Madry et al., 2018) or the KL divergence (Zhang et al., 2019) is used.

For the black-box attack, an adversary generates a dataset $\{\boldsymbol{x}_i, \tilde{y}_i\}_{i=1}^n$ where $\tilde{y}_i$ is an output of a given input $\boldsymbol{x}_i$. Then, the adversary trains a substitute prediction model based on this data set, and generates adversarial examples from the substitute prediction model by PGD (Papernot et al., 2017).

### 2.3 REVIEW OF ADVERSARIAL TRAINING ALGORITHMS

We review some of the adversarial training algorithms which, we think, are related to our proposed algorithm. Typically, adversarial training algorithms consist of the maximization and minimization steps. In the maximization step, we generate adversarial examples for given $\boldsymbol{\theta}$, and in the minimization step, we fix the adversarial examples and update $\boldsymbol{\theta}$. In the followings, we denote $\widehat{\boldsymbol{x}}_i^{\mathrm{pgd}}$ as the adversarial example corresponding to $(\boldsymbol{x}_i, y_i)$ generated by PGD.

#### 2.3.1 ALGORITHMS MINIMIZING THE ROBUST RISK DIRECTLY

**PGD-AT** Madry et al. (2018) proposes PGD-AT which updates $\boldsymbol{\theta}$ by minimizing

$$\sum_{i=1}^n \ell_{\mathrm{ce}}(f_{\boldsymbol{\theta}}(\widehat{\boldsymbol{x}}_i^{\mathrm{pgd}}), y_i),$$

where $\ell_{\mathrm{ce}}$ is the cross-entropy loss.

**GAIR-AT** Geometry Aware Instance Reweighted Adversarial Training (GAIR-AT) (Zhang et al., 2021) is a modification of PGD-AT, where the weighted robust risk is minimized and more weights are given to samples closer to the decision boundary. To be more specific, the weighted empirical risk of GAIR-AT is given as

$$\sum_{i=1}^n w_\theta(\boldsymbol{x}_i, y_i) \ell_{\mathrm{ce}}(f_{\boldsymbol{\theta}}(\widehat{\boldsymbol{x}}_i^{\mathrm{pgd}}), y_i),$$

where $\kappa_\theta(\boldsymbol{x}_i, y_i) = \min\left(\min(\{t : F_{\boldsymbol{\theta}}(\boldsymbol{x}_i^{(t)}) \neq y_i\}), T\right)$ for a prespecified maximum iteration $T$ and $w_\theta(\boldsymbol{x}_i, y_i) = (1 + \tanh(5(1 - 2\kappa_\theta(\boldsymbol{x}_i, y_i)/T)))/2$.

There are other similar modifications of PGA-AT including Max-Margin Adversarial (MMA) Training (Ding et al., 2020) and Friendly Adversarial Training (FAT) (Zhang et al., 2020).

#### 2.3.2 ALGORITHMS MINIMIZING A REGULARIZED EMPIRICAL RISK

Robust risk, natural risk and boundary risk are defined by

$$\begin{aligned}
\mathcal{R}_{\mathrm{rob}}(\boldsymbol{\theta}) &= \mathbb{E}_{(\mathbf{X},Y)} \mathbb{1}\left\{\exists \mathbf{X}' \in \mathcal{B}_p(\mathbf{X}, \varepsilon) : F_{\boldsymbol{\theta}}(\mathbf{X}') \neq Y\right\}, \\
\mathcal{R}_{\mathrm{nat}}(\boldsymbol{\theta}) &= \mathbb{E}_{(\mathbf{X},Y)} \mathbb{1}\left\{F_{\boldsymbol{\theta}}(\mathbf{X}) \neq Y\right\}, \\
\mathcal{R}_{\mathrm{bdy}}(\boldsymbol{\theta}) &= \mathbb{E}_{(\mathbf{X},Y)} \mathbb{1}\left\{\exists \mathbf{X}' \in \mathcal{B}_p(\mathbf{X}, \varepsilon) : F_{\boldsymbol{\theta}}(\mathbf{X}) \neq F_{\boldsymbol{\theta}}(\mathbf{X}'), F_{\boldsymbol{\theta}}(\mathbf{X}) = Y\right\}.
\end{aligned}$$

Zhang et al. (2019) shows

$$\mathcal{R}_{\mathrm{rob}}(\boldsymbol{\theta}) = \mathcal{R}_{\mathrm{nat}}(\boldsymbol{\theta}) + \mathcal{R}_{\mathrm{bdy}}(\boldsymbol{\theta}).$$

By treating $\mathcal{R}_{\mathrm{bdy}}(\boldsymbol{\theta})$ as the regularization term, various regularized risks for adversarial training have been proposed.

**TRADES** Zhang et al. (2019) proposes the following regularized empirical risk which is a surrogate version of the upper bound of the robust risk:

$$\sum_{i=1}^n \left\{\ell_{\mathrm{ce}}(f_\theta(\boldsymbol{x}_i), y_i) + \lambda \cdot \mathrm{KL}(\mathbf{p}_\theta(\cdot|\boldsymbol{x}_i) \| \mathbf{p}_\theta(\cdot|\widehat{\boldsymbol{x}}_i^{\mathrm{pgd}}))\right\},$$

**HAT** Helper based training (Rade & Moosavi-Dezfolli, 2022) is a variation of TRADES where an additional regularization term based on helper examples is added to the regularized risk. The role of helper examples is to restrain the decision boundary from having excessive margins. HAT minimizes the following regularized empirical risk:

$$\sum_{i=1}^{n} \left[ \ell_{\text{ce}} \left( f_{\boldsymbol{\theta}} \left( \boldsymbol{x}_i \right), y_i \right) + \lambda \cdot \text{KL} \left( \mathbf{p}_{\boldsymbol{\theta}} \left( \cdot | \boldsymbol{x}_i \right) \| \mathbf{p}_{\boldsymbol{\theta}} (\cdot | \widehat{\boldsymbol{x}}_i^{\text{pgd}}) \right) + \gamma \ell_{\text{ce}} \left( f_{\boldsymbol{\theta}}(\boldsymbol{x}_i^{\text{helper}}), F_{\boldsymbol{\theta}_{\text{pre}}}(\widehat{\boldsymbol{x}}_i^{\text{pgd}}) \right) \right],$$

where $\boldsymbol{\theta}_{\text{pre}}$ is the parameter of a pre-trained model only with clean examples, $\boldsymbol{x}_i^{\text{helper}} = \boldsymbol{x}_i + 2(\widehat{\boldsymbol{x}}_i^{\text{pgd}} - \boldsymbol{x}_i)$.

**MART** Misclassification Aware adveRsarial Training (MART) (Wang et al., 2020) minimizes

$$\sum_{i=1}^{n} \left\{ \ell_{\text{margin}}(f_{\boldsymbol{\theta}}(\widehat{\boldsymbol{x}}_i^{\text{pgd}}), y_i) + \lambda \cdot \text{KL}(\mathbf{p}_{\boldsymbol{\theta}}(\cdot | \boldsymbol{x}_i) \| \mathbf{p}_{\boldsymbol{\theta}}(\cdot | \widehat{\boldsymbol{x}}_i^{\text{pgd}}))(1 - p_{\boldsymbol{\theta}}(y_i | \boldsymbol{x}_i)) \right\}, \tag{3}$$

where $\ell_{\text{margin}}(f_{\boldsymbol{\theta}}(\widehat{\boldsymbol{x}}_i^{\text{pgd}}), y_i) = -\log p_{\boldsymbol{\theta}}(y_i | \widehat{\boldsymbol{x}}_i^{\text{pgd}}) - \log(1 - \max_{k \neq y_i} p_{\boldsymbol{\theta}}(k | \widehat{\boldsymbol{x}}_i^{\text{pgd}}))$. This objective function can be regarded as the regularized robust risk and thus MART can be considered as a hybrid algorithm of PGD-AT and TRADES.

## 3 ANTI-ROBUST WEIGHTED REGULARIZATION (ARoW)

In this section, we develop a new adversarial training algorithm called Anti-Robust Weighted Regularization (ARoW), which is an algorithm minimizing a regularized risk. We propose a new regularized term which applies more regularization to data vulnerable to adversarial attacks than other existing algorithms such as TRADES and HAT do. Our new regularized term is motivated by the upper bound of the robust risk derived in the following section.

### 3.1 UPPER BOUND OF THE ROBUST RISK

In this subsection, we provide an upper bound of the robust risk for multi-classification problem which is stated in the following theorem. The proof is deferred to Appendix A.

**Theorem 1.** *For a given score function $f_{\boldsymbol{\theta}}$, let $z(\cdot)$ be an any measurable mapping from $\mathcal{X}$ to $\mathcal{X}$ satisfying*

$$z(\boldsymbol{x}) \in \underset{\boldsymbol{x}' \in \mathcal{B}_p(\boldsymbol{x}, \varepsilon)}{\arg \max} \mathbb{1} \left( F_{\boldsymbol{\theta}}(\boldsymbol{x}) \neq F_{\boldsymbol{\theta}}(\boldsymbol{x}') \right).$$

*for every $\boldsymbol{x} \in \mathcal{X}$. Then, we have*

$$\mathcal{R}_{rob}(\boldsymbol{\theta}) \leq \mathbb{E}_{(\mathbf{X}, Y)} \mathbb{1}(Y \neq F_{\boldsymbol{\theta}}(\mathbf{X})) + \mathbb{E}_{(\mathbf{X}, Y)} \mathbb{1}(F_{\boldsymbol{\theta}}(\mathbf{X}) \neq F_{\boldsymbol{\theta}}(z(\mathbf{X}))) \mathbb{1} \left\{ p_{\boldsymbol{\theta}}(Y | z(\mathbf{X})) < 1/2 \right\} \tag{4}$$

The upper bound (4) consists of the two terms : the first term is the natural risk itself and the second term is an upper bound of the boundary risk. This upper bound is motivated by the upper bound derived in TRADES (Zhang et al., 2019). For binary classification problems, Zhang et al. (2019) shows that

$$\mathcal{R}_{\text{rob}}(\boldsymbol{\theta}) \leq \mathbb{E}_{(\mathbf{X}, Y)} \phi(Y f_{\boldsymbol{\theta}}(\mathbf{X})) + \mathbb{E}_{\mathbf{X}} \phi(f_{\boldsymbol{\theta}}(\mathbf{X}) f_{\boldsymbol{\theta}}(z(\mathbf{X}))), \tag{5}$$

where

$$z(\boldsymbol{x}) \in \underset{\boldsymbol{x}' \in \mathcal{B}_p(\boldsymbol{x}, \varepsilon)}{\arg \max} \phi \left( f_{\boldsymbol{\theta}}(\boldsymbol{x}) f_{\boldsymbol{\theta}}(\boldsymbol{x}') \right)$$

and $\phi(\cdot)$ is an upper bound of $\mathbb{1}(\cdot < 0)$. Our upper bound (4) is a modification of the upper bound (5) for multiclass problems where $\phi(\cdot)$ and $f_{\boldsymbol{\theta}}$ in (5) are replaced by $\mathbb{1}(\cdot < 0)$ and $F_{\boldsymbol{\theta}}$, respectively. A key difference, however, between (4) and (5) is the term $\mathbb{1} \{ p_{\boldsymbol{\theta}}(Y | z(\mathbf{X})) < 1/2 \}$ at the last part of (4) that is not in (5).

It is interesting to see that the upper bound in Theorem 1 becomes equal to the robust risk for binary classification problems. That is, the upper bound (4) is an another formulation of the robust risk. However, this rephrased formula of the robust risk is useful since it provides a new learning algorithm when the indicator functions are replaced by their surrogates as we do.

---

**Algorithm 1** ARoW Algorithm

---

**Input** : network $f_{\boldsymbol{\theta}}$, training dataset $\mathcal{D} = \left\{ (\boldsymbol{x}_i, y_i) \in \mathbb{R}^{d+1} : i = 1, \cdots, n \right\}$, learning rate $\eta$,
   hyperparameters $(\lambda, \alpha)$ of (6), number of epochs $T$, number of batch $B$, batch size $K$

**Output** : adversarially robust network $f_{\boldsymbol{\theta}}$

1: **for** $t = 1, \cdots, T$ **do**
2:     **for** $b = 1, \cdots, B$ **do**
3:         $\widehat{\boldsymbol{x}}_{t,b,k}^{\mathrm{pgd}} \leftarrow \underset{\boldsymbol{x}' \in \mathcal{B}_p(\boldsymbol{x}_{t,b,k}, \varepsilon)}{\arg\max} \ \mathrm{KL}(\mathbf{p}_{\boldsymbol{\theta}}(\cdot | \boldsymbol{x}_{t,b,k}) \| \mathbf{p}_{\boldsymbol{\theta}}(\cdot | \boldsymbol{x}')) \ ; \ \boldsymbol{x}_{t,b,k} \in \mathbb{R}^d, k = 1, \ldots, K$
4:         $\boldsymbol{\theta} \leftarrow \boldsymbol{\theta} - \eta \frac{1}{K} \nabla_{\boldsymbol{\theta}} \mathcal{R}_{\mathrm{ARoW}}(\boldsymbol{\theta}; \{(\boldsymbol{x}_{t,b,k}, y_{t,b,k})\}_{k=1}^{K}, \lambda, \alpha)$, where $\mathcal{R}_{\mathrm{ARoW}}$ is (6).
5:     **end for**
6: **end for**
7: **Return** $f_{\boldsymbol{\theta}}$

---

## 3.2 ALGORITHM

By replacing the indicator functions in Theorem 1 by their smooth proxies, we propose a new regularized risk and develop the corresponding adversarial learning algorithm called the Anti-Robust Weighted Regularization (ARoW) algorithm. The four indicator functions in (4) are replaced by

- the adversarial example $z(\boldsymbol{x})$ is replaced by $\widehat{\boldsymbol{x}}^{\mathrm{pgd}}$ obtained by the PGD algorithm with the KL divergence;
- the term $\mathbb{1}(Y \neq F_{\boldsymbol{\theta}}(\mathbf{X}))$ is replaced by the label smooth cross-entropy (Müller et al., 2019) $\ell^{\mathrm{LS}}(f_{\boldsymbol{\theta}}(\boldsymbol{x}), y) = -\boldsymbol{y}_{\alpha}^{\mathrm{LS}\top} \log \mathbf{p}_{\boldsymbol{\theta}}(\cdot | \boldsymbol{x})$ for a given $\alpha > 0$, where $\boldsymbol{y}_{\alpha}^{\mathrm{LS}} = (1 - \alpha)\mathbf{u}_y + \frac{\alpha}{C}\mathbf{1}_C$, $\mathbf{u}_y \in \mathbb{R}^C$ is the one-hot vector whose the $y$-th entry is 1 and $\mathbf{1}_C \in \mathbb{R}^C$ is the vector whose entries are all 1;
- the term $\mathbb{1}(F_{\boldsymbol{\theta}}(\mathbf{X}) \neq F_{\boldsymbol{\theta}}(z(\mathbf{X})))$ is replaced by $\lambda \cdot \mathrm{KL}(\mathbf{p}_{\boldsymbol{\theta}}(\cdot | \mathbf{X}) \| \mathbf{p}_{\boldsymbol{\theta}}(\cdot | \widehat{\mathbf{X}}^{\mathrm{pgd}}))$ for $\lambda > 0$;
- the term $\mathbb{1}\left\{ p_{\boldsymbol{\theta}}(Y | z(\mathbf{X})) < 1/2 \right\}$ is replaced by its convex upper bound $2(1 - p_{\boldsymbol{\theta}}(Y | \widehat{\mathbf{X}}^{\mathrm{pgd}}))$;

to have the following regularized risk for ARoW, which is a smooth surrogate of the upper bound (4),

$$
\mathcal{R}_{\mathrm{ARoW}}(\boldsymbol{\theta}; \{(\boldsymbol{x}_i, y_i)\}_{i=1}^{n}, \lambda)
$$
$$
:= \sum_{i=1}^{n} \left\{ \ell^{\mathrm{LS}}(f_{\boldsymbol{\theta}}(\boldsymbol{x}_i), y_i) + 2\lambda \cdot \mathrm{KL}(\mathbf{p}_{\boldsymbol{\theta}}(\cdot | \boldsymbol{x}_i) \| \mathbf{p}_{\boldsymbol{\theta}}(\cdot | \widehat{\boldsymbol{x}}_i^{\mathrm{pgd}})) \cdot (1 - p_{\boldsymbol{\theta}}(y_i | \widehat{\boldsymbol{x}}_i^{\mathrm{pgd}})) \right\}. \tag{6}
$$

Here, we introduce the regularization parameter $\lambda > 0$ to control the robustness of a trained prediction model to adversarial attacks. That is, the regularized risk (6) can be considered as a smooth surrogate of the regularized robust risk of $\mathcal{R}_{\mathrm{nat}}(\boldsymbol{\theta}) + \lambda \mathcal{R}_{\mathrm{bdy}}(\boldsymbol{\theta})$.

We use the label smoothing cross-entropy as a surrogate for $\mathbb{1}(Y \neq F_{\boldsymbol{\theta}}(\mathbf{X}))$ instead of the standard cross-entropy to estimate the conditional class probabilities $\mathbf{p}_{\boldsymbol{\theta}}(\cdot | \boldsymbol{x})$ more accurately (Müller et al., 2019). The accurate estimation of $\mathbf{p}_{\boldsymbol{\theta}}(\cdot | \boldsymbol{x})$ is important since it is used in the regularization term of ARoW. It is well known that DNNs trained by minimizing the cross-entropy are poorly calibrated (Guo et al., 2017), and so we use the label smoothing cross-entropy technique. We set $\alpha = 0.2$ in our numerical studies for simplicity even if it can be tuned optimally.

The ARoW algorithm, which learns $\boldsymbol{\theta}$ by minimizing $\mathcal{R}_{\mathrm{ARoW}}(\boldsymbol{\theta}; \{(\boldsymbol{x}_i, y_i)\}_{i=1}^{n}, \lambda)$, is summarized in Algorithm 1.

**Comparison to TRADES**   A key difference of the regularized risks of ARoW and TRADES is that TRADES does not have the term $(1 - p_{\boldsymbol{\theta}}(y_i | \widehat{\boldsymbol{x}}_i^{\mathrm{pgd}}))$ at the last part of (6). That is, ARoW puts more regularization to samples which are vulnerable to adversarial attacks (i.e. $p_{\boldsymbol{\theta}}(y_i | \widehat{\boldsymbol{x}}_i^{\mathrm{pgd}})$ is small). Note that this term is motivated by the tighter upper bound of the robust risk (4) and thus is expected to lead better results. Numerical studies confirm that it really works.

**Comparison to MART**   Although the objective function in MART (3) has no theoretical basis, it is similar with the objective function of ARoW. But, there are two main differences. First, the

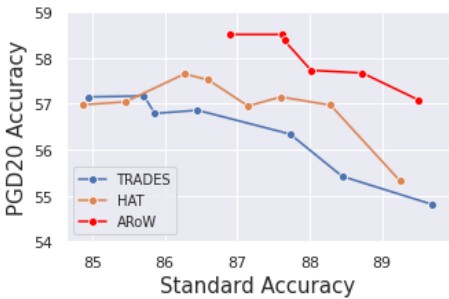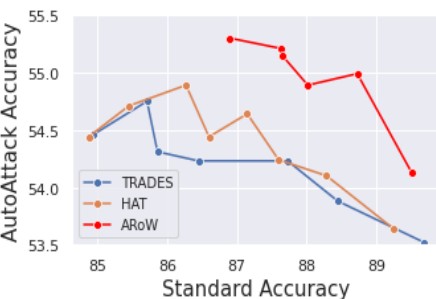

Figure 1: **Comparison of ARoW, TRADES and HAT with varying** $\lambda$. The $x$-axis and $y$-axis are the standard and robust accuracies, respectively. The robust accuracies in the left panel are against PGD$^{20}$ while the robust accuracies in the right panel are against AutoAttack. We exclude the results of MART from the figures because its rboust and standard accuracies are too low.

supervised loss term of ARoW is the label smoothing loss with clean examples, whereas MART uses the margin cross entropy loss with adversarial examples. Second, regularization term in MART is proportional to $(1 - p_{\boldsymbol{\theta}}(y|\boldsymbol{x}))$ while that in ARoW is proportional to $(1 - p_{\boldsymbol{\theta}}(y|\widehat{\boldsymbol{x}}^{\mathrm{pgd}}))$. In the numerical studies, we observe that ARoW outperforms MART with large margins. These would be partly because ARoW is theoretically well motivated.

## 4 EXPERIMENTS

In this section, we investigate the ARoW algorithm in view of robustness and generalization by analyzing the three benchmark data sets - CIFAR10 (Krizhevsky, 2009) , F-MINST (Xiao et al., 2017) and SVHN dataset (Netzer et al., 2011). In particular, we show that ARoW is superior to existing algorithms including TRADES (Zhang et al., 2019), HAT (Rade & Moosavi-Dezfolli, 2022) and MART (Wang et al., 2020) as well as PGD-AT (Madry et al., 2018) and GAIR-AT (Zhang et al., 2021) to achieve state-of-art performances. WideResNet-34-10 (WRN-34-10) (Zagoruyko & Komodakis, 2016) and ResNet-18 (He et al., 2016) are used for CIFAR10 while ResNet-18 (He et al., 2016) is used for F-MNIST and SVHN. Experimental details are presented in Appendix B.

### 4.1 COMPARISON OF ARoW TO TRADES, HAT AND MART

We compare ARoW to the the regularization algorithms TRADES (Zhang et al., 2019), HAT (Rade & Moosavi-Dezfolli, 2022) and MART (Wang et al., 2020) explained in Section 2.3.2. Table 1 shows that ARoW outperforms the other regularization algorithms for various data sets and architectures in terms of both the standard and robust accuracies. The selected values of the hyper-parameters for the other algorithms are listed in Appendix B.2.

To investigate whether ARoW dominates its competitors uniformly with respect to the regularization parameter $\lambda$, we compare the trade-off between the standard and robust accuracies of ARoW and other regularization algorithms when $\lambda$ varies. Figure 1 draws the plots of the standard accuracies in the $x$-axis and the robust accuracies in the $y$-axis obtained by the corresponding algorithms with various values of $\lambda$. For this experiment, we use CIFAR10 and WideResNet-34-10 (WRN-34-10) architecture.

The trade-off between the standard and robust accuracies is well observed (i.e. a larger regularization parameter $\lambda$ yields lower standard accuracy but higher robust accuracy). Moreover, we can clearly see that ARoW uniformly dominates TRADES and HAT (and MART) regardless of the choice of the regularization parameter and the methods for adversarial attack. Additional results for the trade-off are provided in Appendix F.1.

### 4.2 COMPARISON OF ARoW TO PGD-AT AND GAIR-AT

We compare ARoW with PGD-AT (Madry et al., 2018) and GAIR-AT (Zhang et al., 2021) which are the algorithms minimizing the robust risk directly. Table 2 shows that ARoW outperforms PGD-AT and GAIR-AT in terms of the standard accuracy and the robust accuracy against to AutoAttack (Croce & Hein, 2020b). GAIR-AT is, however, better for PGD$^{20}$ attack than ARoW. This would be mainly

Table 1: **Comparison of ARoW, TRADES, HAT and MART.** We conduct the experiment three times with different seeds and present the averages of the accuracies with the standard errors in the brackets.

| Method | CIFAR10 (WRN-34-10) | | | CIFAR10 (ResNet-18) | | |
|---|---|---|---|---|---|---|
| | Stand | PGD$^{20}$ | AA | Stand | PGD$^{20}$ | AA |
| TRADES | 85.86(0.09) | 56.79(0.08) | 54.31(0.08) | 82.41(0.07) | 52.68(0.22) | 49.63(0.25) |
| HAT | 86.98(0.10) | 56.81(0.17) | 54.63(0.07) | 83.05(0.03) | 52.91(0.08) | 49.60(0.02) |
| MART | 84.69(0.18) | 55.67(0.13) | 50.95(0.09) | 74.87(0.95) | 53.68(0.30) | 49.61(0.24) |
| ARoW | **87.65**(0.02) | **58.38**(0.09) | **55.15**(0.14) | **82.53**(0.13) | **55.08**(0.16) | **51.33**(0.18) |
| Method | SVHN (ResNet-18) | | | FMNIST (ResNet-18) | | |
| | Stand | PGD$^{20}$ | AA | Stand | PGD$^{20}$ | AA |
| TRADES | 91.62(0.49) | 58.75(0.19) | 51.06(0.93) | 91.92(0.04) | 88.33(0.03) | 88.19(0.04) |
| HAT | 91.72(0.12) | 58.66(0.06) | 51.67(0.12) | 92.10(0.11) | 88.09(0.16) | 87.93(0.13) |
| MART | 91.64(0.41) | 60.57(0.27) | 49.95(0.42) | 92.14(0.05) | 88.10(0.10) | 87.88(0.14) |
| ARoW | **92.79**(0.24) | **61.14**(0.74) | **51.93**(0.33) | **92.26**(0.05) | **88.73**(0.03) | **88.54**(0.04) |

because of the gradient masking (Papernot et al., 2018; 2017) - PGD does not find an adversarial example well. See Appendix C for details about gradient masking

Table 2: **Comparison of ARoW to PGD-AT and GAIR-AT.** We conduct the experiment three times with different seeds and present the averages of the accuracies with the standard errors in the brackets.

| Method | CIFAR10 (WRN-34-10) | | | CIFAR10 (ResNet-18) | | |
|---|---|---|---|---|---|---|
| | Stand | PGD$^{20}$ | AA | Stand | PGD$^{20}$ | AA |
| PGD-AT | 87.02(0.20) | 57.50(0.12) | 53.98(0.14) | 82.42(0.05) | 53.48(0.11) | 49.30(0.07) |
| GAIR-AT | 85.44(0.10) | 67.27(0.07) | 46.41(0.07) | 81.09(0.12) | 64.89(0.04) | 41.35(0.16) |
| ARoW | **87.65**(0.02) | **58.38**(0.09) | **55.15**(0.14) | **82.53**(0.13) | **55.08**(0.16) | **51.33**(0.18) |
| Method | SVHN (ResNet-18) | | | FMNIST (ResNet-18) | | |
| | Stand | PGD$^{20}$ | AA | Stand | PGD$^{20}$ | AA |
| PGD-AT | 92.75(0.04) | 59.05(0.46) | 47.66(0.52) | 92.25(0.06) | 87.43(0.03) | 87.19(0.03) |
| GAIR-AT | 91.95(0.40) | 70.29(0.18) | 38.26(0.48) | 90.96(0.10) | 87.25(0.01) | 85.00(0.12) |
| ARoW | **92.79**(0.24) | **61.14**(0.74) | **51.93**(0.33) | **92.26**(0.05) | **88.73**(0.03) | **88.54**(0.04) |

## 4.3 ANALYSIS WITH EXTRA DATA

For improving performance on CIFAR10, Carmon et al. (2019) and Rebuffi et al. (2021) use extra unlabeled data sets with TRADES. Carmon et al. (2019) uses an additional subset of 500K extracted from 80 Million Tiny Images (80M-TI) and Rebuffi et al. (2021) uses a data set of 1M synthetic samples generated by a denoising diffusion probabilistic model (DDPM) (Ho et al., 2020) along with the SiLU activation function and Exponential Moving Average (EMA). Further, Rade & Moosavi-Dezfolli (2022) shows that HAT achieves the SOTA performance for these extra data.

Table 3 compares ARoW with the exiting algorithms for extra data, which shows that ARoW achieves the state-of-the-art performance when extra data are available even though the margins compared to HAT are not significant. Note that ARoW has advantages other than the high robust accuracies. For example, ARoW is easy to implement compared to HAT since HAT requires a pre-trained model and it needs additional memory. Moreover, as we will see in Section 4.5, ARoW improves the fairness compared to TRADE while HAT does not.

## 4.4 ABLATION STUDIES

We study the following three issues - (i) the effect of label smoothing to ARoW, (ii) the role of the new regularization term in ARoW to improve robustness and (iii) modifications of ARoW by applying tools which improve existing adversarial training algorithms.

### 4.4.1 EFFECT OF LABEL SMOOTHING

Table 4 indicates that label smoothing is helpful not only for ARoW but also for TRADES. This would be partly because the regularization terms in ARoW and TRADES depend on the conditional

Table 3: **Comparison of ARoW to other adversarial algorithms with extra data on CIFAR10.**

| Model | Extra data | Method | Stand | PGD$^{20}$ | AutoAttack |
|---|---|---|---|---|---|
| WRN-28-10 | 80M-TI(500K) | Carmon et al. (2019) | 89.69 | 62.95 | 59.58 |
| | | Rebuffi et al. (2021) | 90.47 | 63.06 | 60.57 |
| | | HAT | 91.50 | 63.42 | **60.96** |
| | | ARoW | **91.57** | **64.64** | 60.91 |
| ResNet-18 | 80M-TI(500K) | Carmon et al. (2019) | 87.07 | 56.86 | 53.16 |
| | | Rebuffi et al. (2021) | 87.67 | 59.20 | 56.24 |
| | | HAT | 88.98 | 59.29 | 56.40 |
| | | ARoW | **89.04** | **60.38** | **56.54** |
| | DDPM(1M) | Carmon et al. (2019) | 82.61 | 56.16 | 52.82 |
| | | Rebuffi et al. (2021) | 83.46 | 56.89 | 54.22 |
| | | HAT | 86.09 | 58.61 | 55.44 |
| | | ARoW | **86.72** | **59.50** | **55.57** |

class probabilities and it is well known that label smoothing is helpful for the calibration of the conditional class probabilities (Pereyra et al., 2017).

Moreover, the results in Table 4 imply that label smoothing is not a main reason for ARoW to outperform TRADES. Even without label smoothing, ARoW is still superior to TRADES (even with the label smoothing). Appendix F.2 presents the results of an additional experiment to assess the effect of label smoothing to the performance.

Table 4: **Comparison of TRADES and ARoW with/without label smoothing.** With WRN-28-10 architecture and CIFAR10 dataset, we use $\lambda = 6$ for TRADES while use $\lambda = 3$ for ARoW.

| Method | Standard | PGD$^{20}$ | AutoAttack |
|---|---|---|---|
| TRADES w/o-LS | 85.86(0.09) | 56.79(0.08) | 54.31(0.08) |
| TRADES w/-LS | 86.33(0.08) | 57.45(0.02) | 54.66(0.08) |
| ARoW w/o-LS | 86.83(0.16) | 58.34(0.09) | 55.01(0.10) |
| ARoW w/-LS | **87.65**(0.02) | **58.38**(0.09) | **55.15**(0.14) |

#### 4.4.2 ROLE OF THE NEW REGULARIZATION TERM IN ARoW

The regularization term of ARoW puts more regularization to less robust samples, and thus we expect that ARoW improves the robustness of less robust samples much. To confirm this conjecture, we do a small experiment.

First, we divide the test data into four groups - least robust, less robust, robust and highly robust according to the values of $p_{\boldsymbol{\theta}_{\mathrm{PGD}}}(y_i|\widehat{\boldsymbol{x}}_i^{\mathrm{pgd}})$ ($< 0.3$, $0.3 \sim 0.5$, $0.5 \sim 0.7$ and $> 0.7$), where $\boldsymbol{\theta}_{\mathrm{PGD}}$ is the parameter learned by PGD-AT (Madry et al., 2018)[1].

Then, for each group, we check how many samples become robust for ARoW and TRADES, respectively, whose results are presented in Table 5. Note that ARoW improves the robustness of least robust samples most compared with TRADES. We believe that this improvement is due to the regularization term in ARoW that enforces more regularization on less robust samples.

Table 5: **Role of the new regularization term in ARoW.** # **Rob**$_{\mathrm{TRADES}}$ and # **Rob**$_{\mathrm{ARoW}}$ represent the number of samples which are only robust to TRADES but not to ARoW, or vice versa. **Diff.** and **Rate of Impro.** denote (# **Rob**$_{\mathrm{ARoW}}$ - # **Rob**$_{\mathrm{TRADES}}$) and **Diff.** / # **Rob**$_{\mathrm{TRADES}}$)

| Sample's Robustness | # **Rob**$_{\mathrm{TRADES}}$ | # **Rob**$_{\mathrm{ARoW}}$ | Diff. | Rate of Impro. (%) |
|---|---|---|---|---|
| Least Robust | 317 | 357 | 40 | 12.62 |
| Less Robust | 945 | 1008 | 63 | 6.67 |
| Robust | 969 | 1027 | 58 | 5.99 |
| Highly Robust | 3524 | 3529 | 5 | 0.142 |

---

[1]We use PGD-AT instead of a standard non-robust training algorithm since all samples become least robust for a non-robust prediction model.

### 4.4.3 Modifications of ARoW

There are many useful tools which improve existing adversarial training algorithms. Examples are Adversarial Weight Perturbation (AWP) (Wu et al., 2020) and Friendly Adversarial Training (FAT) (Zhang et al., 2020). AWP is a tool to find a flat minimum of the objective function and FAT uses early-stopped PGD when generating adversarial examples in the training phase. Details about AWP and FAT are given in Appendix F.4.

We investigate how ARoW performs when it is modified by such a tool. We consider the two modifications of ARoW - ARoW-AWP and ARoW-FAT, where ARoW-AWP searches a flat minimum of the ARoW objective function and ARoW-FAT uses early-stopped PGD in the training phase of ARoW.

Table 6 compares ARoW-AWP and ARoW-FAT to TRDAES-AWP and TRADES-FAT. Both of AWP and FAT are helpful for ARoW and TRADES but ARoW still outperforms TRADES with large margins even after modified by AWP or FAT.

Table 6: **Modifications of TRADES and ARoW.** We use CIFAR10 dataset and ResNet-18 architecture. More details of hyerparameters are provided in Appendix F.4.

| Method | AWP | | | FAT | | |
|---|---|---|---|---|---|---|
| | Standard | PGD$^{20}$ | AutoAttack | Standard | PGD$^{20}$ | AutoAttack |
| TRADES | 82.10(0.09) | 53.56(0.18) | 49.56(0.23) | 82.96(0.08) | 52.76(0.22) | 49.83(0.28) |
| ARoW | **84.98**(0.11) | **55.55**(0.15) | **50.64**(0.18) | **86.21**(0.06) | **53.37**(0.20) | **50.07**(0.17) |

### 4.5 Improved Fairness

Xu et al. (2021) reports that TRADES (Zhang et al., 2019) increases the variation of the per-class accuracies (accuracy in each class) which is not desirable in view of fairness. In turn, Xu et al. (2021) proposes the Fair-Robust-Learning (FRL) algorithm to alleviate this problem. Even if fairness becomes improved, the standard and robust accuracies of FRL are worse than TRADES.

In contrast, Table 7 shows that ARoW improves the fairness as well as the standard and robust accuracies compared to TRADES. This desirable property of ARoW can be partly understood as follows. The main idea of ARoW is to impose more robust regularization to less robust samples. In turn, samples in less accurate classes tend to be more vulnerable to adversarial attacks. Thus, ARoW improves the robustness of samples in less accurate classes which results in improved robustness as well as improved generalization for such less accurate classes. The class-wise accuracies are presented in Appendix G.

Table 7: **Class-wise accuracy disparity for CIFAR10.** We report the accuracy (ACC), the worst-class accuracy (WC-Acc) and the standard deviation of class-wise accuracies (SD) for each method.

| Method | Standard | | | PGD$^{10}$ | | |
|---|---|---|---|---|---|---|
| | Acc | WC-Acc | SD | Acc | WC-Acc | SD |
| TRADES | 85.69 | 67.10 | 9.27 | 57.38 | 27.10 | 16.97 |
| HAT | 86.74 | 65.40 | 11.12 | 57.92 | 24.20 | 18.26 |
| ARoW | **87.58** | **74.51** | **7.11** | **59.32** | **31.05** | **15.67** |

## 5 Conclusion and Future Works

In this paper, we derived an upper bound of the robust risk and developed a new algorithm for adversarial training called ARoW which minimizes a surrogate version of the derived upper bound. A novel feature of ARoW is to impose more regularization on less robust samples than TRADES. The results of numerical experiments shows that ARoW improves the standard and robust accuracies simultaneously to achieve state-of-the-art performances. In addition, ARoW enhances the fairness of the prediction model without hampering the accuracies.

When we developed a computable surrogate of the upper bound of the robust risk in Theorem 1, we replaced $\mathbb{1}(F_{\boldsymbol{\theta}}(\mathbf{X}) \neq F_{\boldsymbol{\theta}}(z(\mathbf{X})))$ by $\mathrm{KL}(\mathbf{p}_{\boldsymbol{\theta}}(\cdot|\mathbf{X})||\mathbf{p}_{\boldsymbol{\theta}}(\cdot|\widehat{\mathbf{X}}^{\mathrm{pgd}}))$. The KL divergence, however, is

not an upper bound of the 0-1 loss and thus our surrogate is not an upper bound of the robust risk. We employed the KL divergence surrogate to make the objective function of ARoW be similar to that of TRADES. It would be worth pursuing to devise an alternative surrogate for the 0-1 loss to reduce the gap between the theory and algorithm.

We have seen in Section 4.5 that ARoW improves fairness as well as accuracies. The advantage of ARoW in view of fairness is an unexpected by-product, and it would be interesting to develop a more principled way of enhancing the fairness further without hampering the accuracy.

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

# Appendices

## A    PROOF OF THEOREM 1

In this section, we prove Theorem 1. The following lemma provides the key inequality for the proof.

**Lemma 2.** *For a given score function $f_{\boldsymbol{\theta}}$, let $z(\cdot)$ be an any measurable mapping from $\mathcal{X}$ to $\mathcal{X}$ satisfying*

$$z(\boldsymbol{x}) \in \underset{\boldsymbol{x}' \in \mathcal{B}_p(\boldsymbol{x}, \varepsilon)}{\arg\max} \mathbb{1}\left(F_{\boldsymbol{\theta}}(\boldsymbol{x}) \neq F_{\boldsymbol{\theta}}(\boldsymbol{x}')\right)$$

*for every $\boldsymbol{x} \in \mathcal{X}$. Then, we have*

$$\begin{aligned}
\mathbb{1}\left\{\exists \boldsymbol{x}' \in \mathcal{B}_p(\boldsymbol{x}, \varepsilon) : F_{\boldsymbol{\theta}}(\boldsymbol{x}) \neq F_{\boldsymbol{\theta}}(\boldsymbol{x}'), F_{\boldsymbol{\theta}}(\boldsymbol{x}) = Y\right\} \\
\leq \mathbb{1}\left\{F_{\boldsymbol{\theta}}(\boldsymbol{x}) \neq F_{\boldsymbol{\theta}}(z(\boldsymbol{x})), Y \neq F_{\boldsymbol{\theta}}(z(\boldsymbol{x}))\right\}
\end{aligned} \tag{A.1}$$

*Proof.* The inequality holds obviously if $\mathbb{1}\left\{F_{\boldsymbol{\theta}}(\boldsymbol{x}) \neq F_{\boldsymbol{\theta}}(z(\boldsymbol{x})), Y \neq F_{\boldsymbol{\theta}}(z(\boldsymbol{x}))\right\} = 1$. Hence, it suffices to show that $\mathbb{1}\left\{\exists \boldsymbol{x}' \in \mathcal{B}_p(\boldsymbol{x}, \varepsilon) : F_{\boldsymbol{\theta}}(\boldsymbol{x}) \neq F_{\boldsymbol{\theta}}(\boldsymbol{x}'), F_{\boldsymbol{\theta}}(\boldsymbol{x}) = Y\right\} = 0$ when either $F_{\boldsymbol{\theta}}(\boldsymbol{x}) = F_{\boldsymbol{\theta}}(z(\boldsymbol{x}))$ or $Y = F_{\boldsymbol{\theta}}(z(\boldsymbol{x}))$.

Suppose $F_{\boldsymbol{\theta}}(\boldsymbol{x}) = F_{\boldsymbol{\theta}}(z(\boldsymbol{x}))$. It trivially holds that $\mathbb{1}\left(F_{\boldsymbol{\theta}}(\boldsymbol{x}) \neq F_{\boldsymbol{\theta}}(z(\boldsymbol{x}))\right) \leq \mathbb{1}\left(F_{\boldsymbol{\theta}}(\boldsymbol{x}) \neq F_{\boldsymbol{\theta}}(\boldsymbol{x}')\right)$ for every $\boldsymbol{x}' \in \mathcal{X}$ since $\mathbb{1}\left(F_{\boldsymbol{\theta}}(\boldsymbol{x}) \neq F_{\boldsymbol{\theta}}(z(\boldsymbol{x}))\right) = 0$ and the equality holds if and only if $F_{\boldsymbol{\theta}}(z(\boldsymbol{x})) = F_{\boldsymbol{\theta}}(\boldsymbol{x}')$. By the definition of $z(\boldsymbol{x})$, the left side of (A.1) is 0 since $\mathbb{1}\left\{\exists \boldsymbol{x}' \in \mathcal{B}_p(\boldsymbol{x}, \varepsilon) : F_{\boldsymbol{\theta}}(\boldsymbol{x}) \neq F_{\boldsymbol{\theta}}(\boldsymbol{x}')\right\} = 0$, and hence the inequality holds.

Suppose $Y = F_{\boldsymbol{\theta}}(z(\boldsymbol{x}))$. If $F_{\boldsymbol{\theta}}(\boldsymbol{x}) = Y$ and there exists $\boldsymbol{x}'$ in $\mathcal{B}_p(\boldsymbol{x}, \varepsilon)$ such that $F_{\boldsymbol{\theta}}(\boldsymbol{x}') \neq F_{\boldsymbol{\theta}}(\boldsymbol{x})$, then we have $F_{\boldsymbol{\theta}}(\boldsymbol{x}') \neq Y = F_{\boldsymbol{\theta}}(\boldsymbol{x}) = F_{\boldsymbol{\theta}}(z(\boldsymbol{x}))$. In turn, it implies $\mathbb{1}\left(F_{\boldsymbol{\theta}}(\boldsymbol{x}) \neq F_{\boldsymbol{\theta}}(z(\boldsymbol{x}))\right) < \mathbb{1}\left(F_{\boldsymbol{\theta}}(\boldsymbol{x}) \neq F_{\boldsymbol{\theta}}(\boldsymbol{x}')\right)$, which is a contradiction to the definition of $z(\boldsymbol{x})$. Hence, the left side of (A.1) should be 0, and we complete the proof of the inequality. $\square$

**Theorem 1.** *For a given score function $f_{\boldsymbol{\theta}}$, let $z(\cdot)$ be an any measurable mapping from $\mathcal{X}$ to $\mathcal{X}$ satisfying*

$$z(\boldsymbol{x}) \in \underset{\boldsymbol{x}' \in \mathcal{B}_p(\boldsymbol{x}, \varepsilon)}{\arg\max} \mathbb{1}\left(F_{\boldsymbol{\theta}}(\boldsymbol{x}) \neq F_{\boldsymbol{\theta}}(\boldsymbol{x}')\right).$$

*for every $\boldsymbol{x} \in \mathcal{X}$. Then, we have*

$$\mathcal{R}_{rob}(\boldsymbol{\theta}) \leq \mathbb{E}_{(\mathbf{X}, Y)} \mathbb{1}(Y \neq F_{\boldsymbol{\theta}}(\mathbf{X})) + \mathbb{E}_{(\mathbf{X}, Y)} \mathbb{1}(F_{\boldsymbol{\theta}}(\mathbf{X}) \neq F_{\boldsymbol{\theta}}(z(\mathbf{X}))) \mathbb{1}\left\{p_{\boldsymbol{\theta}}(Y|z(\mathbf{X})) < 1/2\right\} \tag{4}$$

*Proof.* Note that $\mathcal{R}_{\mathrm{rob}}(\boldsymbol{\theta}) = \mathcal{R}_{\mathrm{nat}}(\boldsymbol{\theta}) + \mathcal{R}_{\mathrm{bdy}}(\boldsymbol{\theta})$ where $\mathcal{R}_{\mathrm{nat}}(\boldsymbol{\theta}) = \mathbb{E}_{(\mathbf{X}, Y)} \mathbb{1}\left\{F_{\boldsymbol{\theta}}(\mathbf{X}) \neq Y\right\}$ and $\mathcal{R}_{\mathrm{bdy}}(\boldsymbol{\theta}) = \mathbb{E}_{(\mathbf{X}, Y)} \mathbb{1}\left\{\exists \mathbf{X}' \in \mathcal{B}_p(\mathbf{X}, \varepsilon) : F_{\boldsymbol{\theta}}(\mathbf{X}) \neq F_{\boldsymbol{\theta}}(\mathbf{X}'), F_{\boldsymbol{\theta}}(\mathbf{X}) = Y\right\}$.

Since

$$\begin{aligned}
\mathcal{R}_{\mathrm{bdy}}(\boldsymbol{\theta}) &= \mathbb{E}_{(\mathbf{X}, Y)} \mathbb{1}\left\{\exists \mathbf{X}' \in \mathcal{B}_p(\mathbf{X}, \varepsilon) : F_{\boldsymbol{\theta}}(\mathbf{X}) \neq F_{\boldsymbol{\theta}}(\mathbf{X}'), F_{\boldsymbol{\theta}}(\mathbf{X}) = Y\right\} \\
&\leq \mathbb{E}_{(\mathbf{X}, Y)} \mathbb{1}\left\{F_{\boldsymbol{\theta}}(\mathbf{X}) \neq F_{\boldsymbol{\theta}}(z(\mathbf{X})), Y \neq F_{\boldsymbol{\theta}}(z(\mathbf{X}))\right\} (\because \text{ Lemma } 2) \\
&= \mathbb{E}_{(\mathbf{X}, Y)} \mathbb{1}\left\{F_{\boldsymbol{\theta}}(\mathbf{X}) \neq F_{\boldsymbol{\theta}}(z(\mathbf{X}))\right\} \mathbb{1}\left\{Y \neq F_{\boldsymbol{\theta}}(z(\mathbf{X}))\right\} \\
&\leq \mathbb{E}_{(\mathbf{X}, Y)} \mathbb{1}\left\{F_{\boldsymbol{\theta}}(\mathbf{X}) \neq F_{\boldsymbol{\theta}}(z(\mathbf{X}))\right\} \mathbb{1}\left\{p_{\boldsymbol{\theta}}(Y|z(\mathbf{X})) < 1/2\right\},
\end{aligned}$$

the inequality (4) holds. $\square$

## B   DETAILED SETTINGS FOR THE EXPERIMENTS WITH BENCHMARK DATASETS

### B.1   EXPERIMENTAL SETUP

For CIFAR10, SVHN and FMNIST datasets, input images are normalized into [0, 1]. Random crop and random horizontal flip with probability 0.5 are used for CIFAR10 while only random horizontal flip with probability 0.5 is applied for SVHN. For FMNIST, augmentation is not used.

For generating adversarial examples in the training phase, $\text{PGD}^{10}$ with random initial, $p = \infty$, $\varepsilon = 8/255$ and $\nu = 2/255$ is used, where $\text{PGD}^T$ is the output of the PGD algorithm (2) with $T$ iterations. For training prediction models, the SGD with momentum 0.9, weight decay $5 \times 10^{-4}$, the initial learning rate of 0.1 and batch size of 128 is used and the learning rate is reduced by a factor of 10 at 60 and 90 epochs. Stochastic weighting average (SWA) (Izmailov et al., 2018) is employed after 50-epochs for preventing from robust overfitting (Rice et al., 2020) as Chen et al. (2021) does.

For evaluating the robustness in the test phase, $\text{PGD}^{20}$ and AutoAttack are used for adversarial attacks, where AutoAttack consists of three white box attacks - APGD and APGD-DLR in Croce & Hein (2020b) and FAB in Croce & Hein (2020a) and one black box attack - Square Attack (Andriushchenko et al., 2020). To the best of our knowledge, AutoAttack is the strongest attack. The final model is set to be the best model against $\text{PGD}^{10}$ on the test data among those obtained until 120 epochs.

### B.2   HYPERPARAMETER SETTING

Table 8: **Selected hyperparameters.** Hyperparameters used in the numerical studies in Section 4.1 and Section 4.2.

| Dataset | Model | Method | $\lambda$ | $\gamma$ | Weight Decay | $\alpha$ | SWA |
|---------|-------|--------|-----------|----------|--------------|----------|-----|
| CIFAR10 | WRN-34-10 | TRADES | 6 | - | $5e^{-4}$ | - | o |
| | | HAT | 4 | 0.25 | $5e^{-4}$ | - | o |
| | | MART | 6 | - | $2e^{-4}$ | - | x |
| | | PGD-AT | - | - | $5e^{-4}$ | - | o |
| | | GAIR-AT | - | - | $5e^{-4}$ | - | o |
| | | ARoW | 3 | - | $5e^{-4}$ | 0.2 | o |
| | ResNet-18 | TRADES | 6 | - | $5e^{-4}$ | - | o |
| | | HAT | 4 | 0.5 | $5e^{-4}$ | - | o |
| | | MART | 6 | - | $5e^{-4}$ | - | x |
| | | PGD-AT | - | - | $5e^{-4}$ | - | o |
| | | GAIR-AT | - | - | $5e^{-4}$ | - | o |
| | | ARoW | 5 | - | $5e^{-4}$ | 0.2 | o |
| SVHN | ResNet-18 | TRADES | 6 | - | $5e^{-4}$ | - | x |
| | | HAT | 4 | 0.5 | $5e^{-4}$ | - | x |
| | | MART | 6 | - | $5e^{-4}$ | - | x |
| | | PGD-AT | - | - | $5e^{-4}$ | - | x |
| | | GAIR-AT | - | - | $5e^{-4}$ | - | x |
| | | ARoW | 3 | - | $5e^{-4}$ | 0.2 | x |
| FMNIST | ResNet-18 | TRADES | 6 | - | $5e^{-4}$ | - | x |
| | | HAT | 5 | 0.15 | $5e^{-4}$ | - | x |
| | | MART | 6 | - | $5e^{-4}$ | - | x |
| | | PGD-AT | - | - | $5e^{-4}$ | - | x |
| | | GAIR-AT | - | - | $5e^{-4}$ | - | x |
| | | ARoW | 6 | - | $5e^{-4}$ | 0.2 | x |

Table 8 presents the hyperparameters used on our experiments. Most of the hyperparameters are set to be the ones used in the previous studies. The weight decay parameter is set to be $5e^{-4}$ in most experiments, which is the well-known optimal value. Only for MART (Wang et al., 2020) with WRN34-10, we use weight decay $2e^{-4}$ as Wang et al. (2020) did since MART works poorly with $5e^{-4}$. We use stochastic weight averaging (SWA) for CIFAR10 except MART. Note that SWA is not

used in the experiments of Wang et al. (2020), and we confirm that SWA is not helpful for MART. Effects of SWA for all methods are provided in Appendix F.3.

## C CHECKING THE GRADIENT MASKING

Table 9: **Comparison of GAIR-AT and ARoW**. We compare the robustness of GAIR-AT (Zhang et al., 2021) and ARoW against the four attacks used in AutoAttack on CIFAR10. The results are based on WRN-34-10. We set $\lambda = 3$ for ARoW.

| Method | Standard | PGD | APGD | APGD-DLR | FAB | SQUARE |
|---|---|---|---|---|---|---|
| GAIR-AT | 85.44(0.170) | **67.27**(0.07) | **63.14**(0.16) | 46.48(0.07) | 49.35(0.05) | 55.19(0.16) |
| ARoW | **87.65**(0.02) | 58.38(0.09) | 56.07(0.14) | **55.17**(0.11) | **56.69**(0.17) | **63.50**(0.08) |

*Gradient masking* (Papernot et al., 2018; 2017) is the case that the gradient of the loss for a given non-robust datum is almost zero (i.e. $\nabla_{\boldsymbol{x}}\ell_{\text{ce}}(f_{\boldsymbol{\theta}}(\boldsymbol{x}), y) \approx \boldsymbol{0}$). In this case, PGD cannot generate an adversarial example. We can check the ocuurence of gradient masking when a prediction model is robust to the PGD attack but not robust to attacks such as FAB (Croce & Hein, 2020a), APGD-DLR (Croce & Hein, 2020b) and SQUARE (Andriushchenko et al., 2020).

In Table 9, the robustness of GAIR-AT becomes worse much for the three attacks in AutoAttack except APGD (Croce & Hein, 2020b) while the robustness of ARoW remains stable regardless of the adversarial attacks. Since APGD uses the gradient of the loss, this observation implies that the gradient masking occurs in GAIR-AT while it does not in ARoW.

Better performance of GAIR-AT for PGD[20] attack in Table 2 is not because GAIR-AT is robust to adversarial attacks but because adversarial examples obtained by PGD are close to clean samples. This claim is supported by the fact that GAIR-AT performs poorly for AutoAttack while it is still robust to other PGD-based adversarial attacks. Moreover, gradient masking for GAIR-AT is already reported by Hitaj et al. (2021).

## D DETAILED SETTING FOR THE EXPERIMENTS WITH EXTRA DATA

Table 10: **Selected hyperparameters.** Hyperparameters used in the numerical studies in Section 4.3. We do not employ cutmix augmentation (Yun et al., 2019) as does in Rade & Moosavi-Dezfolli (2022).

| Model | Method | $\lambda$ | $\gamma$ | Weight Decay | $\alpha$ | EMA | SiLU |
|---|---|---|---|---|---|---|---|
| WRN-28-10 | Carmon et al. (2019) | 6 | - | $5e^{-4}$ | - | x | x |
| | Rebuffi et al. (2021) | 6 | - | $5e^{-4}$ | - | o | o |
| | HAT | 4 | 0.25 | $5e^{-4}$ | - | o | o |
| | ARoW | 3.5 | - | $5e^{-4}$ | 0.2 | o | o |
| ResNet-18 | Carmon et al. (2019) | 6 | - | $5e^{-4}$ | - | x | x |
| | Rebuffi et al. (2021) | 6 | - | $5e^{-4}$ | - | o | o |
| | HAT | 4 | 0.25 | $5e^{-4}$ | - | o | o |
| | ARoW | 3.5 | - | $5e^{-4}$ | 0.2 | o | o |

In Section 4.3, we presented the results of ARoW on CIFAR10 with extra unlabeled data used in Carmon et al. (2019) and Rebuffi et al. (2021). In this section, we provide experimental details.

Rebuffi et al. (2021) use the SiLU activation function and exponential model averaging (EMA) based on TRADES. For HAT (Rade & Moosavi-Dezfolli, 2022) and ARoW, we use the SiLU activation function and exponential model averaging (EMA) with weight decay factor 0.995 as is done in Rebuffi et al. (2021). The cosine annealing learning rate scheduler (Loshchilov & Hutter, 2017) is used with the batch size 512. The final model is set to be the best model against PGD[10] on the test data among those obtained until 500 epochs.

# E ADDITIONAL RESULTS WITH EXTRA DATA

## E.1 RESULTS ON CIFAR 100

For additional experiments, we analyzed CIFAR-100 in WRN-34-10 and found that ARoW still outperforms the other competitors.

Table 11: **Comparison of ARoW to competitors on CIFAR100.** We compare ARoW to PGD-AT, TRADES, HAT and MART on CIFAR100. We used WRN-28-10 architecture.

| Method | Stand | PGD | AutoAttack |
|--------|-------|-------|------------|
| PGD-AT | 62.20 | 32.27 | 28.66 |
| TRADES | 62.23 | 33.45 | 29.07 |
| HAT | 60.42 | 33.75 | 29.42 |
| MART | 59.76 | 33.37 | 29.68 |
| ARoW | **62.38** | **34.74** | **30.42** |

## E.2 ADDITIONAL RESULTS WITH EXTRA DATA

In the main manuscript, we use architecture of ResNet18, while Rade & Moosavi-Dezfolli (2022) use PreAct-ResNet18. For better comparison, we conduct an additional experiment with extra data where the same architecture - PreaAct-ResNet18 is used. In addition, we set batch size to 1024 which is used in Rade & Moosavi-Dezfolli (2022). Table 12 shows that ARoW outperforms HAT both on standard accuracy(+0.29%) and robust accuracy(+0.11%) against autoattack.

Table 12: **Performance with extra data (Carmon et al.) on CIFAR10.** We brought the values in the paper as reported in Rade & Moosavi-Dezfolli (2022).

| Method | Standard | AutoAttack |
|--------|----------|------------|
| HAT | 89.02 | 57.67 |
| ARoW | 89.31 | 57.78 |

# F ABLATION STUDY

## F.1 THE TRADE-OFF DUE TO THE CHOICE OF $\lambda$

Table 13 presents the trade-off between the generalization and robustness accuracies of ARoW on CIFAR10 due to the choice of $\lambda$, where ResNet18 is used. The trade-off is obviously observed.

Table 13: **Standard and robust accuracies of ARoW on CIFAR10 for varying $\lambda$.**

| $\lambda$ | Standard | $PGD^{20}$ | AutoAttack |
|-----------|----------|------------|------------|
| TRADES($\lambda = 6$) | 82.41 | 52.68 | 49.63 |
| ARoW($\lambda = 2.5$) | 85.30 | 53.80 | 49.66 |
| ARoW($\lambda = 3.0$) | 84.65 | 54.23 | 50.11 |
| ARoW($\lambda = 3.5$) | 83.86 | 54.13 | 50.15 |
| ARoW($\lambda = 4.0$) | 83.73 | 54.20 | 50.55 |
| ARoW($\lambda = 4.5$) | 82.97 | 54.69 | 50.83 |
| ARoW($\lambda = 5.0$) | 82.53 | 55.08 | 51.33 |

## F.2 THE EFFECT OF LABEL SMOOTHING

Table 14 presents the standard and robust accuracies of ARoW on CIFAR10 for various values of the smoothing parameter $\alpha$ in the label smoothing where the regularization parameter $\lambda$ is fixed at 3 and ResNet18 is used.

Table 14: **Standard and robust accuracies of ARoW on CIFAR10 for varying $\alpha$.**

| $\alpha$ | Standard | PGD$^{20}$ | AutoAttack |
|------|----------|-----------|------------|
| 0.05 | 83.54 | 53.10 | 49.88 |
| 0.10 | 84.10 | 53.29 | 49.75 |
| 0.15 | 84.36 | 53.56 | 49.67 |
| 0.20 | 84.52 | 53.68 | 49.96 |
| 0.25 | 84.48 | 53.53 | 49.93 |
| 0.30 | 84.55 | 53.53 | 49.89 |
| 0.35 | 84.66 | 54.19 | 50.03 |
| 0.40 | 84.65 | 54.23 | 50.11 |

## F.3 EFFECT OF STOCHASTIC WEIGHT AVERAGING (SWA)

We compare the standard and robust accuracies of the adversarial training algorithms with and without SWA whose results are summarized in Table 15. SWA improves the accuracies for all the algorithms except MART. Without SWA, ARoW is competitive to HAT, which is known to be the SOTA method. However, ARoW dominates HAT when SWA is applied.

Table 15: **Effects of SWA on CIFAR10 with WideResNet 34-10.** We conduct the experiment three times with different seeds and present the averages of the accuracies with the standard errors in the brackets. 'w/o' stands for 'without'.

| | Method | Standard | PGD$^{20}$ | AutoAttack |
|---------|--------|--------------|--------------|--------------|
| | TRADES | 85.86(0.09) | 56.79(0.08) | 54.31(0.08) |
| | HAT | 86.98(0.10) | 56.81(0.17) | 54.63(0.07) |
| SWA | MART | 78.41(0.07) | 56.04(0.09) | 48.94(0.09) |
| | PGD-AT | 87.02(0.20) | 57.50(0.12) | 53.98(0.14) |
| | ARoW | **87.59**(0.02) | **58.61**(0.09) | **55.21**(0.14) |
| | TRADES | 85.48(0.12) | 56.06(0.08) | 53.16(0.17) |
| | HAT | 87.53(0.02) | 56.41(0.09) | **53.38**(0.10) |
| w/o-SWA | MART | 84.69(0.18) | 55.67(0.13) | 50.95(0.09) |
| | PGD-AT | 86.88(0.09) | 54.15(0.16) | 51.35(0.14) |
| | ARoW | **87.60**(0.02) | **56.47**(0.10) | 52.95(0.06) |

## F.4 AWP AND FAT

### F.4.1 ADVERSARIAL WEIGHT PERTURBATION (AWP)

For a given objective function of the adversarial training, AWP (Wu et al., 2020) tries to find a flat minimum in the parameter space. Wu et al. (2020) proposes TRADES-AWP, which minimizes

$$\min_{\boldsymbol{\theta}} \max_{\|\boldsymbol{\delta}_l\| \le \gamma \|\boldsymbol{\theta}_l\|} \frac{1}{n} \sum_{i=1}^{n} \left\{ \ell_{\mathrm{ce}}(f_{\boldsymbol{\theta}+\boldsymbol{\delta}}(\boldsymbol{x}_i), y_i) + \lambda \cdot \mathrm{KL}(\mathbf{p}_{\boldsymbol{\theta}+\boldsymbol{\delta}}(\cdot|\boldsymbol{x}_i) \| \mathbf{p}_{\boldsymbol{\theta}+\boldsymbol{\delta}}(\cdot|\widehat{\boldsymbol{x}}_i^{\mathrm{pgd}})) \right\},$$

where $\boldsymbol{\theta}_l$ is the weight vector of $l$-th layer and $\gamma$ is the weight perturbation size. Inspired by TRADES-AWP, we propose ARoW-AWP which minimizes

$$\min_{\boldsymbol{\theta}} \max_{\|\boldsymbol{\delta}_l\| \le \gamma \|\boldsymbol{\theta}_l\|} \frac{1}{n} \sum_{i=1}^{n} \left\{ \ell_{\mathrm{ce}}(f_{\boldsymbol{\theta}+\boldsymbol{\delta}}(\boldsymbol{x}_i), y_i) \right.$$
$$\left. + 2\lambda \cdot \mathrm{KL}(\mathbf{p}_{\boldsymbol{\theta}+\boldsymbol{\delta}}(\cdot|\boldsymbol{x}_i) \| \mathbf{p}_{\boldsymbol{\theta}+\boldsymbol{\delta}}(\cdot|\widehat{\boldsymbol{x}}_i^{\mathrm{pgd}})) \cdot (1 - p_{\boldsymbol{\theta}}(y_i|\widehat{\boldsymbol{x}}_i^{\mathrm{pgd}})) \right\}.$$

In our experiment, we set $\gamma$ to be 0.005 which is the value used in Wu et al. (2020) and do not use SWA as did in original paper.

### F.4.2 FRIENDLY ADVERSARIAL TRAINING (FAT)

Zhang et al. (2020) suggests early-stopped PGD which uses a data-adaptive iterations of PGD when an adversarial example is generated. TRADES-FAT, which uses the early-stopped PGD in TRADES, minimizes

$$\sum_{i=1}^{n} \ell_{\mathrm{ce}}(f_{\boldsymbol{\theta}}(\boldsymbol{x}_i), y_i) + \lambda \cdot \mathrm{KL}(\mathbf{p}_{\theta}(\cdot|\boldsymbol{x}_i) \| \mathbf{p}_{\theta}(\cdot|\widehat{\boldsymbol{x}}_i^{(t_i)}))$$

where $t_i = \min\left\{\min\{t : F_{\boldsymbol{\theta}}(\widehat{\boldsymbol{x}}_i^{(t)}) \neq y_i\} + K, T\right\}$. Here, $T$ is the maximum iterations of PGD.

We propose an adversarial training algorithm ARoW-FAT by combining ARoW and early-stopped PGD. ARoW-FAT minimizes the following regularized empirical risk:

$$\sum_{i=1}^{n} \left\{ \ell_{\alpha}^{\mathrm{LS}}(f_{\theta}(\boldsymbol{x}_i), y_i) + 2\lambda \cdot \mathrm{KL}(\mathbf{p}_{\theta}(\cdot|\boldsymbol{x}_i) \| \mathbf{p}_{\theta}(\cdot|\widehat{\boldsymbol{x}}_i^{(t_i)})) \cdot (1 - p_{\boldsymbol{\theta}}(y_i|\widehat{\boldsymbol{x}}_i^{(t_i)})) \right\}.$$

In the experiments, we set $K$ to be 2, which is the value used in Zhang et al. (2020).

## G IMPROVED FAIRNESS

Table 16: **Comparison of per-class robustness and generalization of TRADES and ARoW.** $\mathbf{Rob}_{\mathrm{TRADES}}$ and $\mathbf{Rob}_{\mathrm{ARoW}}$ are the robust accuracies against $\mathrm{PGD}^{20}$ of TRADES and ARoW, respectively. $\mathbf{Stand}_{\mathrm{TRADES}}$ and $\mathbf{Stand}_{\mathrm{ARoW}}$ are the standard accuracies.

| Class | $\mathbf{Rob}_{\mathrm{TRADES}}$ | $\mathbf{Rob}_{\mathrm{ARoW}}$ | $\mathbf{Stand}_{\mathrm{TRADES}}$ | $\mathbf{Stand}_{\mathrm{ARoW}}$ |
|---|---|---|---|---|
| 0(Airplane) | 64.8 | 66.7 | 88.3 | 91.6 |
| 1(Automobile) | 77.5 | 77.5 | 93.7 | 95.3 |
| 2(Bird) | 38.5 | 43.1 | 72.5 | 80.6 |
| 3(Cat) | 26.1 | 30.2 | 65.9 | 75.1 |
| 4(Deer) | 35.6 | 40.3 | 83.4 | 87.5 |
| 5(Dog) | 48.6 | 47.2 | 76.0 | 79.3 |
| 6(Frog) | 67.8 | 63.6 | 94.2 | 95.2 |
| 7(Horse) | 69.7 | 69.3 | 91.0 | 92.7 |
| 8(Ship) | 62.3 | 70.1 | 90.9 | 94.9 |
| 9(Truck) | 75.3 | 76.3 | 93.5 | 93.5 |

In Table 16, we present the per-class robust and standard accuracies of the prediction models trained by TRADES and ARoW. We can see that ARoW is highly effective for classes difficult to be classified such as Bird, Cat, Deer and Dog. For such classes, ARoW improves much not only the standard accuracies but also the robust accuracies. For example, in the class 'Cat', which is the most difficult class (the lowest standard accuarcy for TRADES and ARoW), the robustness and generalization are improved by 4.1 percentage point ($26.1\% \rightarrow 30.2\%$) and 9.2 percentage point ($65.9\% \rightarrow 75.1\%$) by ARoW compared with TRADES, respectively. This desirable results would be mainly due to the new regularization term in ARoW. Usually, difficult classes are less robust to adversarial attacks. By putting more regularization on less robust classes, ARoW improves the accuracies of less robust classes more.

