# OpenReview forum: "Improving Adversarial Robustness by Putting More Regularizations on Less Robust Samples"
_ICLR.cc/2023/Conference — Submitted to ICLR 2023_

### Official Review · Reviewer_m348 · 2022-10-19

**Confidence:** 4
**Clarity, Quality, Novelty And Reproducibility:** The manuscript is clear.
**Correctness:** 2
**Technical Novelty And Significance:** 2
**Empirical Novelty And Significance:** 2
**Recommendation:** 5

**Strength And Weaknesses:**

Strength:
1. Well-written
2. Claim both theoretical and empirical contributions

Weakness:
1. **(1)** I guess Equation A.1 should be '$=$' and Throem 1 (and the proof) should be '$=$' (rather than '$\le$') for **binary classification problems**. Hence, Throem 1 is just another expression of 'R_rob = R_nat + R_bdy'. The second term in Theorm 1 is just another expression of boundary error rather than the upper bound of boundary error.
**(2)** For **multi-classification problems**, I guess Equation A.1 is still $=$ rather than $\le$ as $x'$ includes $z(x)$. **Could authors clarify why use $\le$ rather than $=$ here and what's the meaning of Lemma 2?**
**(3)** I also think the last line of last equation in Page 13 is $=$ (rather than $\le$) for binary classification problems, and $\le$ for **multi-classification problem** is meaningless, for me it seems like **loosing the equation and artificially creating a very loose bound** . I.e., use **($Y\ne F(z(x))$ is the sufficient and unnecessary condition of $p(Y|z(x))<0.5$ for multi-classification problem)** to bound the equation. This is the only technical part in this theorem, could we call this a new theorem and claim this a theoretical contribution (theorem 1 is even a simple expansion equation for binary case)?
**(4) In conclusion, for me, Theorem 1 seems like an equation for binary classification problem and an artificially-created loose bound for multi-classification problem. That is,
$R_{bdy}=\mathbb{E}[\mathbb{1}(F(X)\ne F(z(X)))\mathbb{1}(Y\ne F(z(X)))] \le \mathbb{E}[\mathbb{1}(F(X)\ne F(z(X))) \mathbb{1} (p(Y|z(x))<0.5)]$, which uses a loose and simple bound to bound the original equation. In my opinion, the $\le$ here is the only technical part in Theorem 1.
I cannot get the hidden meaning and theoretical contribution of this bound and think it is essentially meaningless. Could authors clarify what is the specific and profound meaning of Theorem 1?**
**Please point out my mistakes if I understand it incorrectly.**
2. In Sec. 3.2, the term 1 {pθ(Y |z(X)) < 1/2} is replaced by its convex upper bound 2(1 − pθ(Y | \hat X pgd)). This upper bound is loose, especially in a multi-classification problem (like CIFAR-10 in the experiments), this bound is meaningless as it is too loose.
**Thus, for me, the theoretical part in this manuscript seems like a created loose bound plus a created loose bound.**
3. Weak experiments and only a little improvement, need more empirical resuts (e.g., CIFAR-100, compare with AWP-TRADES)


**Summary Of The Paper:**

This work tries to develop an upper bound of the robust risk and design a new algorithm for
adversarial training called ARoW which minimizes a surrogate version of the developed upper bound.

**Summary Of The Review:**

I believe this paper is not good enough to publish in ICLR for the following reasons.
1. (main) Overclaim in theoretical part. Ref weakness 1, 2. I do not think it has sufficient theoretical contribution and its theorem can support the method.
2. (secondary) Weak experiments.

---

> ### Author Response · Authors · 2022-11-12
> **Response to reviewer m348 [Part 2]**
>
> (4) In conclusion, for me, Theorem 1 seems like an equation for binary classification problem and an artificially-created loose bound for multi-classification problem. That is,
>             $R_{bdy} =E_{({\bf X}, Y)} 1 \\{ F_{{\bf \theta}}({\bf X}) \neq F_{{\bf \theta}}(z({\bf X})) \\} 1\\{ Y \neq F_{{\bf  \theta}}(z({\bf X})) \\}
>             \leq  E_{({\bf X}, Y)} 1 \\{ F_{{\bf \theta}}({\bf X}) \neq F_{{\bf \theta}}(z({\bf X})) \\} 1 \\{ p_{{\bf \theta}}(Y | z({\bf X})) < 1/2 \\}$
>         , which uses a loose and simple bound to bound the original equation. In my opinion, the  here is the only technical part in Theorem 1.
>         I cannot get the hidden meaning and theoretical contribution of this bound and think it is essentially meaningless. Could authors clarify what is the specific and profound meaning of Theorem 1?
>         Please point out my mistakes if I understand it incorrectly.
>
> -   As we replied for the previous comment,
>    our aim is not to derive a tightest upper bound. Instead, we tried to improve upper bound used in TRADES to develop an improved adversarial learning algorithm and we succeeded it. That is, the role of Theorem 1 is to provide
>             a theoretical motivation of our adversarial learning algorithm even though the bound of Theorem 1 may not be sufficiently tight (even so, it's tighter than that in TRADES).
>
> - We want to emphasize the importance of theoretical motivations. In a certain sense,
>   our algorithm looks similar to MART. The both algorithms multiply the 1- conditional class probability to the KL divergence in their objective functions. However, our algorithm uses
>             $p_\theta(y|\hat{{\bf x}}^{\text{pgd}})$
>             while MART uses $p_\theta(y|{\bf x}).$
>             The empirical risk of MART is heuristically designed.
>             Our empirical results, which show that MART is much inferior to our algorithm, would suggest that using $p_\theta(y|\hat{{\bf x}}^{\text{pgd}})$ is a right way.
>
> 2. In Sec. 3.2, the term $1 \\{p_{\theta}(Y |z(X)) < 1/2\\}$ is replaced by its convex upper bound $2(1 - p_{\theta}(Y | \hat{x}^\text{pgd}))$. This upper bound is loose, especially in a multi-classification problem (like CIFAR-10 in the experiments), this bound is meaningless as it is too loose. Thus, for me, the theoretical part in this manuscript seems like a created loose bound plus a created loose bound.
>
> - Let $r := p_{{\bf \theta}}(Y | Z({\bf X}))$.
>    We use the inequality $1 \\{r < 1/2\\} \leq 2(1-r)$, which has a same spirit to the hinge loss as a convex upper bound of the 0-1 loss in classification.
>    Since hinge loss is considered as the tightest convex upper bound of 0-1 loss, we do not think that the inequality $1 \\{r < 1/2\\} \leq 2(1-r)$ is loose.
>
> -  In addition, the main theme of our work is not to derive a tight
>    upper bound. Instead, the mission is to develop a good adversarial learning algorithm. By using $1 \\{r < 1/2\\} \leq 2(1-r),$
>     we succeeded in completing our mission.
>
> -  Tightness would be a subjective opinion unless the bound is sharp. The sharp bound of the robust risk is the robust risk itself but any trial to minimize the robust risk directly could fail due to bad local minima. The crucial point in developing a good learning algorithm is not to derive a tighter bound but to derive a good surrogate loss which provides a good (local) minimum.
>
> 3. Weak experiments and only a little improvement, need more empirical results (e.g., CIFAR-100, compare with AWP-TRADES).
>
> - For additional experiments, we analyzed CIFAR-100 in WRN-34-10 and found that ARoW still outperforms the other competitors.  See Table below.
> We will add this table and experimental setting in the revised version.
> \\begin{array}{|c|c|c|}
> \\hline  & \text{Stand} & \text{PGD}^{20} & \text{AutoAttack} \\\\\\hline
>   \text{PGD-AT}  & 62.20 & 32.27 & 28.66  \\\\\\hline
>   \text{TRADES} & 62.23 & 33.45 & 29.07  \\\\\\hline
>   \text{MART}  & 59.76 & 33.37 & 29.68 \\\\\\hline
>   \text{HAT}  & 60.42 & 33.75 & 29.42 \\\\\\hline
>   \text{ARoW}  & 62.83 & 34.74 & 30.42 \\\\\\hline
> \\end{array}
>
> - In Table 6 of the main manuscript, we already compared TRADES-AWP with ARoW-AWP. ARoW-AWP outperforms TRADES-AWP by 2.27\% , 1.99\% and 1.08\%  points for standard accuracy, robust accuracies against PGD-20 and Autoattack, respectively.

---

> ### Author Response · Authors · 2022-11-12
> **Response to reviewer m348 [Part 1]**
>
> We highly appreciate the reviewers for the careful reviews and rich feedback.
>
> $\textbf{Weakness}$
>
> 1. (1) I guess Equation A.1 should be '=' and Throem 1 (and the proof) should be '=' (rather than '$\leq$') for binary classification problems. Hence, Throem 1 is just another expression of 'R_rob = R_nat + R_bdy'. The second term in Theorm 1 is just another expression of boundary error rather than the upper bound of boundary error.
>
> - Yes, you are right. Thank you for pointing out this equivalence.
>   But, the $\leq$ is necessary for multi-classification problems.
>
> - An additional comment: The equivalence of the
> $R_{nat}+R_{bdy}$ and our bound for binary classification problem is also interesting.
> Since our bound uses the term $p_\theta(Y|z(\mathbf{X}))$ which is smooth in $\theta.$ This smoothness is helpful when we replace the upper bound by its smooth surrogate.
>
> (2) For multi-classification problems, I guess Equation A.1 is still $=$ rather than $\leq$ as $x'$ includes $z(x)$. Could authors clarify why use $\leq$ rather than $=$ here and what's the meaning of Lemma 2?
>
> - For multi-classification problems, $\leq$ is necessary. For exmaple,
>             consider a case where $K \geq 3,$  $F_{\bf{\theta}}(\mathbf{x})=1 , F_{\bf{\theta}}(z(\mathbf{x}))=2$ and  $Y=3.$
>              Then, the left hand side of (A.1) is 0 while
>              the right hand side of (A.1) is 1.
>
> (3) I also think the last line of last equation in Page 13 is $=$ (rather than $\leq$) for binary classification problems, and $\leq$  for multi-classification problem is meaningless, for me it seems like loosing the equation and artificially creating a very loose bound . I.e., use ($Y \neq F(z(\mathbf{x}))$ is the sufficient and unnecessary condition of $p(Y|z(x)) < 0.5 $  for multi-classification problem) to bound the equation. This is the only technical part in this theorem, could we call this a new theorem and claim this a theoretical contribution (theorem 1 is even a simple expansion equation for binary case)?
>
> -  As we explained in the reply to the first comment, for binary classifications, every $\leq$ can be replaced by $=.$
>   But, for multi-classification problems, we need $\leq$ since
>
>   $$
>   \qquad     1\\{ Y \neq F_{{\bf \theta}}(z({\bf X})) \\} \leq  1 \\{ p_{{\bf \theta}}(Y | z({\bf X})) < 1/2 \\}.
>   $$
>
> - One may think that the above inequality is too loose and we do not object it. However, our aim is not to derive a tightest upper bound. Instead, we tried to improve upper bound used in TRADES to
>           develop an improved adversarial learning algorithm. $\textbf{The term  }  {\bf \\{ p_{{\bf \theta}}(Y | z({\bf X})) < 1/2 \\}}$, $\textbf{ which makes our bound is tighter }$ $\textbf{than that in TRADES, is one of the main contribution of our work.}$
>
> - Note that
> $$
>   \qquad 1\\{ Y \neq F_{{\bf  \theta}}(z({\bf  X})) \\} \leq  1 \\{ p_{{\bf \theta}}(Y | z({\bf X})) < \alpha \\}
> $$
>
> holds for any $\alpha \geq 1/2$ and we chose $\alpha=1/2$ to make the bound as tight as possible. If $\alpha=1,$ our bound becomes equivalent to that of TRADES.

---

> > ### Comment · Reviewer_m348 · 2022-11-12
> > **Fast reply to 1.(2).**
> >
> > ''**For multi-classification problems,  $\le$ is necessary. For exmaple, consider a case where  $K\ge 3$, $F(x)=1$, $F(z(x))=2$ and $Y=3$. Then, the left hand side of (A.1) is 0 while the right hand side of (A.1) is 1.**''
> >
> > Thanks for reply, I guess it raises new problem, i.e., what is the clear definition of $z(x)=\arg\max_{x'}\mathbb{1}(F(x)\ne F(x'))$?
> >
> > For the case $K= 3$, $F(x)=1$ and $Y=3$. If there exists many $x'$ satisfy $F(x')=2, 3$, does all of these $x'$ satisfy the definition of $z(x)$?
> > **For a specific input $x$**, $z(x)$ is a set, distribution, random variable or constant?
> > If $z(x)$ is not a constant, how to clearly define F(z(x)) and $F(z(x))\ne Y$?
> > If $z(x)$ is a constant, how to compute it If there exists many $x'$ satisfy $F(x')=2, 3$?
> > How can Theory 1 and Lemma 2 rigorously hold? Under which specific definition of $z(x)$?
> >
> > I will reply others later.

---

> > > ### Author Response · Authors · 2022-11-13
> > > **Response to "Fast reply to 1.(2)."**
> > >
> > > Thank you for careful reading.
> > >
> > > 1. What is the clear definition of $z({\bf x}) = \text{argmax}_{{\bf x}^\prime} 1(F({\bf x}) \neq F({\bf x}^\prime))$?
> > > For the case $K=3$, $F({\bf x})=1$ and $Y=3$. If there exists many ${\bf x}^{\prime}$ satisfy $F({\bf x}^{\prime})=2,3$, does all of these ${\bf x}^{\prime}$ satisfy the definition of $z({\bf x})$?
> > >
> > > -  Rigorously, $z$ is any (measurable) mapping from $\mathcal{X}$ to $\mathcal{X}$ satisfying
> > > $$
> > > z({\bf x}) \in \text{argmax}_ {{\bf x}' \in B_{p}({\bf x}, \varepsilon)} 1 ( F_{{\bf \theta}}({\bf x}) \neq F_{\mathbf{\theta}}({\bf x}')).
> > > $$
> > >     Your concern is about the uniqueness  of $z,$ and as you pointed out, $z$
> > >     is not uniquely defined. But, it is not difficult to check that Lemma 2 and Theorem 1 are satisfied for all $z.$
> > >
> > > - When $F_\theta(z(\mathbf{x}))=Y,$ Lemma 2 is still true but  $\leq$ can be replaced by $=$ as pointed by you.
> > > However, when $F_\theta(z(\mathbf{x})) \ne Y,$ we need $\leq$ in Lemma 2.
> > >
> > > 2. If $z({\bf x})$ is a constant, how to compute it if there exists many ${\bf x}^{\prime}$ satisfy $F({\bf x}^{\prime}) = 2,3$?
> > >
> > > - As we explained, $z$ is a mapping (i.e. $z({\bf x})$ is a constant) but not uniquely defined. However, Lemma 2 and Theorem 1 hold for any nonuniquely defined $z.$
> > >
> > >  - In practice, we do not compute $z$ because it is almost impossible. Instead, we
> > >         replace the indicator function in the definition of $z$ by its smooth surrogate and
> > >         compute $z$ by use of the smooth surrogate and gradient ascent algorithm.

---

> > > > ### Comment · Reviewer_m348 · 2022-11-13
> > > > **Response to Response to "Fast reply to 1.(2)."**
> > > >
> > > > If I understand it correctly, $z(\cdot)$ is a one to multi mapping from $\mathcal{X}$ to $\mathcal{X}$. Thus for a specific input $x$, $z(x)$ is not uniquely defined, what is the **specific and clear definition** of $z(x) and F(z(x))$, random variable or set or distribution or others?
> > > >
> > > > By the way, what does $\mathbb{1} [F(x)\ne F(z(x)), Y\ne F(z(x))]$ mean in Eq A.1 when $F(z(x))=2,3$ and $F(x)=1$, $Y=3$?
> > > >
> > > > Could authors present rigorous mathmatical symbols and formulas? In the current manuscript, I am confused about the definitions and formulas, and still think it is not clear and rigorous enough to be published.
> > > >
> > > > What's the difference between $z(x)$ and $[\exists x', F(x)\ne F(x')]$, $[\forall x', F(x)\ne F(x')]$?
> > > >
> > > > ''$z(\cdot)$ is a mapping (i.e. $z(x)$ is a constant) but not uniquely defined.'' Does it mean for a specific $x$, $z(x)$ is **a constant but not uniquely defined**? Can authors define it in math? Sorry for my shallow mathematic knowledge, I have not heard it before and can not understand ''a constant but not uniquely defined''.
> > > >
> > > > **I am also happy to hear what other reviewers and AC think about this.**

---

> > > > > ### Author Response · Authors · 2022-11-14
> > > > > **Response to Response to "Fast reply to 1.(2)."**
> > > > >
> > > > > We are sorry for the confusion due to our vague definition of $z$.
> > > > >     $z(\cdot)$ is a mapping(=function) from $\mathcal{X}$ to $\mathcal{X}$ satisfying
> > > > >  $$
> > > > > z({\bf x}) \in \text{argmax}_ {{\bf x}' \in B_{p}({\bf x}, \varepsilon)} 1 ( F_{{\bf \theta}}({\bf x}) \neq F_{\mathbf{\theta}}({\bf x}')).
> > > > > $$
> > > > >  Even though there are many such $z(\cdot)$s, for a given mapping $z(\cdot)$ and input $x$, $z(x)$ is a uniquely defined element in $\mathcal{X}.$
> > > > >
> > > > > "$z(\cdot)$ is a mapping (i.e. z(x) is a constant) but not uniquely defined." : it means that
> > > > >
> > > > > (1) $z(\cdot)$ is any mapping which satisfy it's definition.
> > > > > (2) for a specific mapping $z(\cdot)$ and input $x$, $z(x)$ is a constant (and uniquely defined).
> > > > >
> > > > > As we explained, Theorem 1 and Lemma 2 are satisfied for all $z(\cdot)$ which satisfy the definition.
> > > > >
> > > > > Rigorously, we can restate Theorem 1 as follows:
> > > > >
> > > > >
> > > > > ---
> > > > > For a given score function $f_{{\theta}},$ let $z(\cdot)$ be an any measurable mapping from $\mathcal{X}$ to $\mathcal{X}$ satisfying
> > > > >     $$
> > > > > z({\bf x}) \in \text{argmax}_ {{\bf x}' \in B_{p}({\bf x}, \varepsilon)} 1 ( F_{{\bf \theta}}({\bf x}) \neq F_{{\bf \theta}}({\bf x}'))
> > > > > $$
> > > > >
> > > > > for every $x \in \mathcal{X}$. Then, we have
> > > > >
> > > > > $$
> > > > > \mathcal{R}_{\text{rob}}({\bf \theta}) \leq {E}\_{({\bf X},Y)} 1 (Y \neq F\_{\bf \theta}({\bf X})) +
> > > > > E\_{({\bf X}, Y)}1(F\_{{\bf \theta}}({\bf X}) \neq F\_{{\bf \theta}} (z({\bf X})) 1 \\{ p\_{{\bf \theta}}(Y | z({\bf X})) < 1/2 \\}.
> > > > > $$
> > > > > ----
> > > > > Also, Lemma 2 can be written similarly.
> > > > > The manuscript has also been revised according to your comment.

---

> > > > > > ### Comment · Reviewer_m348 · 2022-11-14
> > > > > > **Response to Response to "Fast reply to 1.(2)."**
> > > > > >
> > > > > > (1). I think the current mathematical statement is still not rigorous (let us discuss mathematical symbols firstly, the theorem can be discussed only based on correct symbols and formulas).
> > > > > > ''$z(\cdot)$ is **a** mapping (=function) from $\mathcal{X}$ to $\mathcal{X}$ satisfying..., there are many such $z(\cdot)$s''
> > > > > > Thus $z(x)$ can be $x_1', x_2'$..., my question is:
> > > > > > **For a given $x$, what is the mathematical property of $z(x)$? Do you mean $z(x)$ is a constant randomly from satisfied x'? Can you use $\exists$ or $\forall$ to define $z(x)$?**
> > > > > > **Then, for a given dataset $X$, what is the mathematical property of $E_{(X,Y)} 1[F(X)\ne F(z(X)), F(z(X))\ne Y]$? Do you mean it is also a constant based on randomly selected $z(x)$?**
> > > > > >
> > > > > > **Anyway, I think the current mathematical statement is incorrect, and still cannot understand the precise definition of $z(x)$ according to your mathematical formula.**
> > > > > >
> > > > > > (2) According to my guess,  can Theorem1 be **concisely** written as  (remove the mystery $z(x)$) the following?
> > > > > > For multi-classification problem:
> > > > > > $R_{rob}=R_{nat}+R_{bdy}$
> > > > > > $R_{bdy}=E_{(X,Y)} 1[\exists X':F(X)\ne F(X'), F(X)=Y]$
> > > > > > $\quad \quad \le E_{(X,Y)} 1[\exists X':F(X)\ne F(X'), F(X')\ne Y]$
> > > > > > $\quad \quad \le E_{(X,Y)} 1[\exists X':F(X)\ne F(X'), p(Y|X')<0.5]$

---

> > > > > > > ### Author Response · Authors · 2022-11-14
> > > > > > > **Response to Response to "Fast reply to 1.(2)."**
> > > > > > >
> > > > > > > (1)
> > > > > > > A more rigorous statement of Theorem 1 is as follows:
> > > > > > >
> > > > > > > Define the function class $\mathcal{Z}$ as
> > > > > > >
> > > > > > > $$
> > > > > > > \mathcal{Z} := \\{ z(\cdot) : \mathcal{X} \to \mathcal{X} \text{ s.t. } \forall x \in \mathcal{X}, z(x) \in B_p (x,\epsilon) \text{ and } I(F_{\theta} (x) \neq F_{\theta} (z(x)))    = \max_{x^{\prime} \in B_p (x,\epsilon)} I(F_{\theta} (x) \neq F_{\theta} (x^{\prime}))   \\}
> > > > > > > $$
> > > > > > >
> > > > > > > Then, $\forall z(\cdot) \in \mathcal{Z}$, we have
> > > > > > > $$
> > > > > > > \mathcal{R}_{\text{rob}}({\bf \theta}) \leq {E}\_{({\bf X},Y)} 1 (Y \neq F\_{\bf \theta}({\bf X})) +
> > > > > > > E\_{({\bf X}, Y)}1(F\_{{\bf \theta}}({\bf X}) \neq F\_{{\bf \theta}} (z({\bf X})) 1 \\{ p\_{{\bf \theta}}(Y | z({\bf X})) < 1/2 \\}.
> > > > > > > $$
> > > > > > >
> > > > > > > $\textbf{Remark}$: Note that if a function $z(\cdot)$ and an input x are given, then $z(x)$ is unique.
> > > > > > > However, $z(\cdot),$ as a function, is not unique.
> > > > > > >
> > > > > > > (2) Please, read the reply to the first comment.

---

> > > > > > > > ### Comment · Reviewer_m348 · 2022-11-14
> > > > > > > > **Response to Response to "Fast reply to 1.(2)."**
> > > > > > > >
> > > > > > > > Then, let's look at Theory 1:
> > > > > > > > $R_{rob}=R_{nat}+R_{bdy}$  (1)
> > > > > > > > $R_{bdy}=E_{(X,Y)} 1[\exists X':F(X)\ne F(X'), F(X)=Y]$  (2)
> > > > > > > > $\quad \quad = E_{(X,Y)} 1[F(X)\ne F(z(X)), F(X)= Y]$  (3)
> > > > > > > > $\quad \quad \le E_{(X,Y)} 1[F(X)\ne F(z(X)), F(z(X))\ne Y]$  (4)
> > > > > > > > $\quad \quad \le E_{(X,Y)} 1[F(X)\ne F(z(X)), p(Y|z(X))<0.5]$   (5)
> > > > > > > >
> > > > > > > > 1. For binary classification problem, all $\le$ are $=$, it is just the same Eq. (1) from TRADES. While the bound of TRADES is defined in  binary classification problem, so I think the bound in this work can only compare with TRADES for binary classification problem.
> > > > > > > > The claim in this work **''We derive an upper bound of the robust risk which is tighter than that of TRADES''** is an overclaim,  for binary classification problem, **this work rewrote Eq. (1) from TRADES and claimed this rewritten equation is tighter than the bound of TRADES**?
> > > > > > > > It is obvious that **''the equation tighter than the bound''** as TRADES developed the bound based on Eq. (1) and this work just **rewrote** Eq. (1) for binary classification problem.
> > > > > > > >
> > > > > > > > 2. All $\le$ in Theorem 1 come from **multi**-classification problem, I donot think, for multi-classification problem,
> > > > > > > > ((4)->(5): $F(z(X))\ne Y$ $\rightarrow$  $p(Y|z(X))<0.5$) and ((3)->(4): $F(X)= Y$ $\rightarrow$  $F(z(X))\ne Y$) have theoretical contribution and Theorem 1 can be claimed as a **new theorem**.
> > > > > > > >
> > > > > > > > Next, let's look at Sec. 3.2:
> > > > > > > > I still think **''the term $1 [p(Y |z(X)) < 1/2]$ is replaced by its convex upper bound $2(1 − p(Y |X_{pgd}))$''** looses the bound in multi-classification problem.
> > > > > > > > E.g., CIFAR-100, if $p=0.1$, this bound is $1<1.8$.
> > > > > > > >
> > > > > > > > In conclusion, I still believe the theoretical part in this work is **meaningless** (rewrite Eq. (1) from TRADES, easily amplify the equation in multi-classification problem).
> > > > > > > > I cannot get the meaning of Theorem 1, does it mean in  multi-classification problem, if there exists $z(X)$, the area of $p(Y|z(X))<0.5$ is larger than $F(z(X))\ne Y$ and the area of $F(z(X))\ne Y$ is larger than $F(X)= Y$?
> > > > > > > > For me,  the theoretical part in this work still seems like a created loose bound plus a created loose bound, and cannot support the method.

---

> > > > > > > > > ### Comment · Reviewer_hmqr · 2022-11-16
> > > > > > > > > **Comment on Theoretical Analysis**
> > > > > > > > >
> > > > > > > > > First, I would like to thank both the reviewer m348 and the authors for their discussion. Next, I am going to express my feedback on the running concern.
> > > > > > > > >
> > > > > > > > > *Regarding the results in Theorem 1*:
> > > > > > > > > Indeed, the analysis lead to Theorem 1 is not mathematically involved. However, this does not lower its practical value. The theoretical results in this paper does not guarantee any robustness improvements. However, the derived regularizer that is "mathematically inspired" from Theorem 1 raises the value of its contribution. Thus, I do *not* think that the theoretical part in this work is meaningless. In fact, the simplicity of the result increases its value in my opinion.
> > > > > > > > >
> > > > > > > > > *Regarding the tightness and relation of Theorem 1 to TRADES*:
> > > > > > > > > Indeed, the inequality in Theorem 1 is solely reasoned to the multi-classification problem. This should be stated in the main paper as the binary classification case can be reduced to the results in TRADES. However, since the analysis in TRADES was limited to the binary classification problem, this extension is also of value as demonstrated by the effectiveness of the proposed regularizer. Further, I agree that indeed the bound proposed in Theorem 1 is loose. However, and as dictated in the response of the authors, this is the tightest bound.
> > > > > > > > >  I would suggest tuning down the first contribution in this work as it is correct only for the multi-class classification problem. I would also suggest stating the relation between Theorem 1 and TRADES explicitly in both the binary and multi-class classification problems.

---

> > > > > > > > > > ### Comment · Reviewer_m348 · 2022-11-17
> > > > > > > > > > **Response to 'Comment on Theoretical Analysis'**
> > > > > > > > > >
> > > > > > > > > > Thanks, Reviewer hmqr, I understand we may have different subjective opinions for the same objective thing. Thus, in the following, I will provide my subjective opinions based on the objective content of the manuscript.
> > > > > > > > > >
> > > > > > > > > > 1. Why do I think Theorem 1 is meaningless and cannot be compared with TRADES?
> > > > > > > > > > **(objective)** Theorem 1 rewrites Eq. (1) from TRADES, relax the binary classification assumption to multi-classification assumption, then Eq. (1) can be (easily) rewritten to the bound in Theorem 1. (ref https://openreview.net/forum?id=-SBZ8c356Oc&noteId=82Rsk1FRZ7)
> > > > > > > > > > **(subjective)** My concern is, as a **Theorem**, it is meaningless. That is, I do not think it has enough theoretical contribution to be claimed as a **new Theorem**.
> > > > > > > > > > **(subjective)** I think it cannot be compared with the bound of TRADES, as the bound of TRADES comes from Eq. (1) for binary classification problem. To some extent, the bound in this manuscript is just Eq. (1) itself.
> > > > > > > > > >
> > > > > > > > > > 2. Why do I think Theoretical part cannot support the method?
> > > > > > > > > > **(objective)** The different definitions in the theoretical part and the practical method: $z(x)=\arg\max_{x'} \mathbb{1} [F(x)\ne F(x')]$ in theory, $z(x)=\arg\max_{x'} KL [p(x) || p(x')]$ in practice.
> > > > > > > > > > **(subjective)** The first strange definition seems like to be designed to suit Theorem 1 for multi-classification problem. The second definition is normally used in (the theory and practical method of) TRADES. I think Theory 1 does not hold under the practical definition $z(x)=\arg\max_{x'} KL [p(x) || p(x')]$.
> > > > > > > > > > **(objective)**  Eqs. (4) (5) and $2(1-p(Y|X_{pgd}))$ loose the bound (ref https://openreview.net/forum?id=-SBZ8c356Oc&noteId=82Rsk1FRZ7).
> > > > > > > > > > **(subjective)**  So many differences between 'theory' and practical method, I think 'mathematically inspired' is also somewhat 'farfetched'.
> > > > > > > > > >
> > > > > > > > > > Anyway, the above is just my own opinion, I respect everyone's opinion.

---

> > > > > > > > > > > ### Comment · Reviewer_hmqr · 2022-11-17
> > > > > > > > > > > **Response to Reviewer m348**
> > > > > > > > > > >
> > > > > > > > > > > I would like to thank reviewer m348 for the fast response. Here is my response to both points raised in the previous comment:
> > > > > > > > > > >
> > > > > > > > > > > - Regarding the comparison with TRADES:
> > > > > > > > > > > Indeed, Theorem 1 extends the results from TRADES to the multi-classification problem and bounds one term in it. This extension might seem to be easy and simple for reviewer m348. However, this result does not exist in prior work making it at least a novel result to my (subjective) taste. Further, while this (simple) modification might provide doubts about its usefulness, the experiments showed the proposed regularizer is effective.
> > > > > > > > > > > While I might agree that the (simple) extension over TRADES results might not qualify the result in this paper to be a Theorem, I cannot draw the line on when a theoretical result can qualify as a Theorem.
> > > > > > > > > > >
> > > > > > > > > > > - Regarding the differences between theory and practice.
> > > > > > > > > > > The key value of Theorem 1 is proposing the the reqularization in Eq.(6). The main discrepancy between the definition of $z(x)$ and its smooth approximation though KL-Divergence is a common practice as the indicator function is not differentiable. That is, one should ideally optimize directly for an adversary that changes the prediction through $z(x)$. However, this optimization is hard and as a surrogate, the common practice is to use either the standard cross entropy as in adversarial training, or KL divergence as in TRADES and MART.
> > > > > > > > > > >
> > > > > > > > > > > I appreciate the time and effort put from reviewer m348 for this fruitful discussion.

---

> > > > > > > > > > > > ### Comment · Reviewer_m348 · 2022-11-17
> > > > > > > > > > > > **Thanks for Reply, Reviewer hmqr**
> > > > > > > > > > > >
> > > > > > > > > > > > Thanks for your discussion, Reviewer hmqr.
> > > > > > > > > > > >
> > > > > > > > > > > > I think we have no divarications for the objective issues of this manuscript. I insist my subjective opinion, whereas you have your consideration.
> > > > > > > > > > > >
> > > > > > > > > > > > I will be happy to hear from AC and other reviewers.

---

> > > > > > > > > > > > ### Author Response · Authors · 2022-11-18
> > > > > > > > > > > > **Thanks for your effort.**
> > > > > > > > > > > >
> > > > > > > > > > > > Thank you very much for summarizing the arguments between TRADES and ours. We fully agree
> > > > > > > > > > > > with you, and we have revised the paper to reflect fully your advice!

---

> > > > > > > > > > > ### Author Response · Authors · 2022-11-18
> > > > > > > > > > > **Thanks for discussion.**
> > > > > > > > > > >
> > > > > > > > > > > Thank you for your careful and helpful discussions about the upper bounds. We agree that our upper bound is not better than that of TRDAES for binary classification problems. In the revision, we focused sole on multi-class classification problems.  Also, We mentioned that our bound is equal to the robust error for  binary classification problems.

---

> ### Comment · Reviewer_m348 · 2022-12-03
> **Post-Rebuttal**
>
> Thanks All,
>
> I see authors' effort to make their theoretical claim more precise in the final revised manuscript, but also think it could be improved.
> All in all, I am on the fence for this paper.
>
> Reviewer m348

---

### Official Review · Reviewer_cZx4 · 2022-10-23

**Confidence:** 5
**Correctness:** 3
**Technical Novelty And Significance:** 3
**Empirical Novelty And Significance:** 2
**Recommendation:** 6

**Clarity, Quality, Novelty And Reproducibility:**

Clarity: the paper is well written and clearly presents the method and the results.

Quality: the proposed method is well justified, and the set of experiments is reasonable.

Novelty: the modification to the TRADES loss is quite small, but relevant.

Reproducibility: sufficient experimental details and code are provided.

**Strength And Weaknesses:**

Strengths
- The proposed modification of the TRADES loss, while small, is theoretically justified. The paper clearly presents the new scheme and its differences to existing algorithms.

- The experimental results support ARoW in comparison to existing methods. Most of the relevant baselines are included, and several ablation studies are added to analyze the effect of different components of the training algorithm e.g. label smoothing.

- The paper is well written, and it clearly presents the new method, the baselines and the experimental setup. The experiments include different architectures and datasets, and the case of using extra data for training.

Weaknesses
- The main concern is about the results reported for the baselines, especially for the case of additional data. For example, in Table 3, for the case of ResNet-18, HAT attains 56.40% and 55.44% of robust accuracy with the 500k extra images from 80M-TI and DDPM synthetic images respectively, while it is reported to get, for the same setups, 57.67% and 57.09% on [RobustBench](https://robustbench.github.io/index.html). Similarly, for WRN-28-10 with extra data, the model from [A] achieves 62.76%	of robust accuracy, higher than any method in Table 3. Then, it is not clear whether the baselines are optimally tuned.

- In general, the improvements over TRADES and HAT in terms of robustness are quite small, although consistent.

[A] https://arxiv.org/abs/2010.03593

**Summary Of The Paper:**

The paper proposes a modification of TRADES, one of the most popular algorithms to obtain adversarially robust classifiers, to improve its performance: in particular, the regularization term to achieve robustness is weighted to penalize more the training examples which are less robust. In the experimental evaluation on several datasets, the proposed method, ARoW, is shown to achieve better robustness that existing methods, while preserving higher standard accuracy.

**Summary Of The Review:**

The proposed method is reasonable and shows promising results. However, clarifications about the discrepancy of the results for some baselines to the original ones are needed, as well as adding the missing baseline.

---
Update after rebuttal

Given the additional results and clarifications provided during the rebuttal, I increase the initial score to 6.

---

> ### Author Response · Authors · 2022-11-12
> **Response to reviewer cZx4**
>
> We highly appreciate the reviewers for the careful reviews and rich feedback.
>
> $\bf{Weakness}$
>
> 1. The main concern is about the results reported for the baselines, especially for the case of additional data. For example, in Table 3, for the case of ResNet-18, HAT attains 56.40\% and 55.44\% of robust accuracy with the 500k extra images from 80M-TI and DDPM synthetic images respectively, while it is reported to get, for the same setups, 57.67\% and 57.09\% on RobustBench. Similarly, for WRN-28-10 with extra data, the model from [A] achieves 62.76\% of robust accuracy, higher than any method in Table 3. Then, it is not clear whether the baselines are optimally tuned.
>
> [A] https://arxiv.org/abs/2010.03593
> - There are differences in the experimental settings of HAT on RobustBench and [A] with ours.
> - First, we use the batch size 512 due to the limitation of computational resource, while 1024 is used in HAT and [A]. However,  Table 11 in [A] shows that changing the batch size from 512 to 1024 can improve the robust accuracy against AA about 0.7\% points.
> - Second, [A] uses the regenerated extra data which are different from our extra data  released by [1].  [A] shows that replacing the extra data from [1] by the regenerated data improves the robust accuracy against Autoattack by about 0.7\%. Unfortunately, the regenerated data of [A] is not publicly available.
> - Third, we use the cosine learning rate scheduler with 400 total epochs as Carmon [1] did, while [A] and HAT use the multi-step learning scheduler with total 800 total epochs and cyclic learning rate scheduler with total 400 epochs and Nesterov momentum, respectively. Table 6 in [A] shows that changing learning scheduler from cosine to multi-step can improve the robust accuracy against AA about 0.6\% points.
> - Finally, we use the architecture of the ResNet18 structure, while PreActResNet18 is used in HAT [2].
> - For these reasons, the reported performances of the baselines in Table 3 of our main manuscript may differ from the results in the original paper. Since [A] uses the TRADES algorithm and ARoW outperforms TRADES in our setting, we think that ARoW still outperforms TRADES under the experimental setting of [A].
>
> It would be great to provide numerical results for the setting of [A], but we are afraid that it is not possible due to time limitation of rebuttal.
> If possible, we will try to report certain experimental results which supprot that ARoW works well even under the experimental setting of [A] by the end of the discussion period.
>
> [1] Unlabeled Data Improves Adversarial Robustness, In Neurips, 2019.
>
> [2] Reducing Excessive Margin to Achieve a Better Accuracy vs. Robustness Trade-off, In ICLR, 2022.
>
>
>
> 2. In general, the improvements over TRADES and HAT in terms of robustness are quite small, although consistent.
> - Even though the robustness improvements are marginal, the improvements of generalization (i.e. the standard accuracies) are large.
> - As is well known, the regularization parameter $\lambda$ controls the trade-off of generalization and robustness.
> - Since our generalization performance is significantly higher than that of the competitors, we can improves the robustness further at the expense of the generalization performance.
> - For illustration, as shown in the table below, increasing $\lambda$ from 2.5 to 5 can improves robustness significantly  while generalization performance of ARoW is still better than TRADES. The experiments are conducted in ResNet-18 on CIFAR10.
>
> \\begin{array}{|c|c|c|}
> \\hline  & \text{Stand} & \text{PGD}^{20} & \text{AutoAttack} \\\\\\hline
>   \text{TRADES}(\lambda=6) & 82.41 & 52.68 & 49.63  \\\\\\hline
>   \text{ARoW}(\lambda=2.5)  & 85.30 & 53.80 & 49.66 \\\\\\hline
>   \text{ARoW}(\lambda=3.0)  & 84.65 & 54.23 & 50.11 \\\\\\hline
>   \text{ARoW}(\lambda=3.5)  & 83.86 & 54.13 & 50.15 \\\\\\hline
>   \text{ARoW}(\lambda=4.0)  & 83.73 & 54.20 & 50.55 \\\\\\hline
>   \text{ARoW}(\lambda=4.5)  & 82.97 & 54.69 & 50.83 \\\\\\hline
>   \text{ARoW}(\lambda=5.0)  & 82.53 & 55.08 & 51.33 \\\\\\hline
> \\end{array}
>
> - Also, in Figure 1 (WRN-34-10) of the main manuscript, we can see that robust accuracies of ARoW is much higher than those of the competitors  under the same standard accuracy.

---

> > ### Comment · Reviewer_cZx4 · 2022-11-15
> > **Response after rebuttal**
> >
> > I thank the authors for the detailed reply and additional experiments.
> >
> > - I think the additional results varying $\lambda$ should be included (at least the value which attains the highest robustness) in the main part,  since the paper aims at improving robustness rather than trade-off (as in the title).
> >
> > - The results for the case of extra data are still not very convincing: first, looking at Table 3 HAT and ARoW have very close results in all cases, for both clean and robust accuracy. Second, HAT is shown in [2] to achieve 62.50% robust accuracy with WRN-28-10 and the same 500k additional images: since in this case the training images should be the same, it should be possible to compare ARoW in the setup of HAT. In fact, it should be possible to overcome the other differences in the training scheme mentioned e.g. changing the learning rate schedule, the type of RN-18 and batch size (with gradient accumulation if necessary). At the moment it seems that ARoW has no advantage over HAT if extra data is available, which weakens the proposed method.
> >
> > - About the theoretical contribution discussed with Reviewer m348, it seems that the comparison to TRADES should be better presented, especially that the novelty is only in the case of multi-class problems, and the claims adjusted accordingly.
> >
> > Overall, the difference to existing algorithms is quite small, then it seems necessary for the experimental evaluation to clearly supported the proposed method.

---

> > > ### Author Response · Authors · 2022-11-18
> > > **Response to "Response after rebuttal"**
> > >
> > > Thanks for the response.
> > >
> > > 1. I think the additional results varying $\lambda$ should be included (at least the value which attains the highest robustness) in the main part}, since the paper aims at improving robustness rather than trade-off (as in the title).
> > > - Following your comment, we chose $\lambda$ in ARoW that maximizes the robust accuracy and
> > >     revised Tables 1 and 2 accordingly. Still, ARoW outperforms the other algorithms.
> > >     In addition, We  included additional results for varying $\lambda$ on CIFAR 10 with ResNet-18
> > >     in Appendix F.1. to demonstrate the trade-off between robust and standard accuracies.
> > >
> > > 2. The results for the case of extra data are still not very convincing: first, looking at Table 3 HAT and ARoW have very close results in all cases, for both clean and robust accuracy. Second, HAT is shown in [2] to achieve $62.50\%$ robust accuracy with WRN-28-10 and the same 500k additional images: since in this case the training images should be the same, it should be possible to compare ARoW in the setup of HAT. In fact, it should be possible to overcome the other differences in the training scheme mentioned e.g. changing the learning rate schedule, the type of RN-18 and batch size (with gradient accumulation if necessary). At the moment it seems that ARoW has no advantage over HAT if extra data is available, which weakens the proposed method.
> > > - Thank you very much for letting use know the gradient accumulation.
> > >     By applying the gradient accumulation, we were able to obtain the results in the almost same setting to HAT.  As shown in the table below, ARoW outperforms HAT both on standard accuracy($+0.29\\%$) and robust accuracy($+0.11\\%$) against autoattack. We added those results in Appendix E.
> > >     We are doing an experiment with the extra data generated by DDPM and expect that
> > >     the results are available before the due date of the discussion period.
> > >     We will post the results soon, if possible.
> > >
> > > \\begin{array}{|c|c|}
> > > \\hline  & \text{Stand} &  \text{AutoAttack} \\\\\\hline
> > >   \text{HAT} & 89.02 & 57.67   \\\\\\hline
> > >   \text{ARoW}   & 89.31 & 57.78 \\\\\\hline
> > > \\end{array}
> > > - $\textbf{(About ARoW's advantage over HAT if extra data is available)}$
> > >
> > >     + We agree that the improvement of ARoW over HAT is marginal when extra data are available.
> > >     However, ARoW outperforms HAT without extra data. See Figure 1 and Table 1 for the empirical results.
> > >
> > >     + Beside robust accuracies, ARoW has many other advantages compared to HAT.
> > >     + ARoW is much easy to implement compared to HAT since HAT requires a pre-trained model, which needs additional memory.
> > >     + Another advantage of our algorithm is the improvement of fairness on class-wise robustness, which does not occur for HAT (Table 7 in manuscript). Fairness is an important issue for adversarial learning because
> > >     unfair model becomes more vulnerable to adversarial attack when an adversary focuses on
> > >     vulnerable classes only.
> > >
> > > 3. About the theoretical contribution discussed with Reviewer m348, it seems that the comparison to TRADES should be better presented, especially that the novelty is only in the case of multi-class problems, and the claims adjusted accordingly.
> > > - Thank you for your suggestion. We revised the manuscript accordingly.
> > >
> > >
> > > 4. Overall, the difference to existing algorithms is quite small, then it seems necessary for the experimental evaluation to clearly supported the proposed method.
> > > - Even though deciding whether improvements are marginal or significant is rather subjective,
> > >       one thing clear is that  ARoW consistently outperforms HAT without extra data. See Figure 1 and Table 1 for the empirical evidence. Even with extra data, ARoW performs better than HAT even if
> > >       the margins may be thought to be large. In addition, as we explained in the previous answer,
> > >       ARoW has advantages other than the robust accuracies such as easy implementation and improved fairness.

---

> > > ### Author Response · Authors · 2022-11-18
> > > **The result with extra data generated by DDPM.**
> > >
> > > We present the result with extra data generated by DDPM.
> > >
> > > \\begin{array}{|c|c|}
> > > \\hline  & \text{Stand} &  \text{AutoAttack} \\\\\\hline
> > >   \text{HAT}       & 86.86 & 57.09   \\\\\\hline
> > >   \text{ARoW}   & 87.13 & 57.15  \\\\\\hline
> > > \\end{array}

---

> > > > ### Comment · Reviewer_cZx4 · 2022-11-21
> > > > **Further response**
> > > >
> > > > I thank the authors for the additional experiments. I think these results confirm that ARoW over HAT perform similarly with extra data. However, I think the further experiments provided during the rebuttal show that ARoW can outperform the competitors in the standard setup, hence I will increase my score. I'd invite the authors to clearly describe in the text the theoretical novelty (comparison to TRADES) and experimental improvements to the baselines (with and without extra data).

---

### Official Review · Reviewer_hmqr · 2022-11-01

**Confidence:** 4
**Correctness:** 3
**Technical Novelty And Significance:** 3
**Empirical Novelty And Significance:** 3
**Recommendation:** 8

**Clarity, Quality, Novelty And Reproducibility:**

The paper clearly states its motivation, contributions, and places itself within prior art.

**Strength And Weaknesses:**

This appear has several strengths:

- The paper is is well motivated and the added regularizer is theoretically inspired.

- The experimental analysis show the consistent improvement of the proposed method over previous art.

- The paper is well-written. Further, the contributions of this work is placed properly within the literature.

- The wide broad of the empirical results shown in this paper covers many interesting aspects such as combining the proposed approach with AWP, and increasing the fairness.


There are few weaknesses that I hope to be addressed during the discussion:

- While the paper is generally well-written, there few parts that require small adjustments. For example,
In caption of figure 1: “ We exclude MART from the figures because its performance is too bad”

- Generally, the robustness improvements that ARoW provide is marginal. Would the proposed method improve the state-of-the-art model from [A]?

[A] Uncovering the Limits of Adversarial Training against Norm-Bounded Adversarial Examples, 2021.

**Summary Of The Paper:**

This work proposes a simple, effective, and theoretically inspired regularization to enhance the robustness of DNNs agains adversarial attacks.
Extensive experimental results were carried out showing the effectiveness of the proposed approach in providing robustness enhancements.

**Summary Of The Review:**

There several aspects that I like about this work such as the theoretical motivation, the extensive experimental evaluation, and the writing.
However, there are two concerns that  I hope to be addressed in the discussion period.

---

> ### Author Response · Authors · 2022-11-12
> **Response to reviewer hmqr**
>
> We highly appreciate the reviewer hmqr for the careful review and positive feedback.
>
> $\bf{\text{Weakness}}$
>
> 1. While the paper is generally well-written, there few parts that require small adjustments. For example, In caption of figure 1: “ We exclude MART from the figures because its performance is too bad”.
> -  Thank you for your careful reading. We will correct such errors in the revised version.
>
> 2-1. Generally, the robustness improvements that ARoW provided is marginal.
> -  Even though the robustness improvements are marginal, the improvements of generalization (i.e. the standard accuracies) are large.           -  As is well known, the regularization parameter $\lambda$ controls the trade-off of generalization and robustness.
> -  Since our generalization performance is significantly higher than that of the competitors, we can improve the robustness further at the expense of the generalization performance.
> - For illustration, as shown in the table below, increasing $\lambda$ from 2.5 to 5 can improve robustness significantly  while generalization performance of ARoW is still better than TRADES. The experiments are conducted in ResNet-18 on CIFAR10.
>
> \\begin{array}{|c|c|c|}
> \\hline  & \text{Stand} & \text{PGD}^{20} & \text{AutoAttack} \\\\\\hline
>   \text{TRADES}(\lambda=6) & 82.41 & 52.68 & 49.63  \\\\\\hline
>   \text{ARoW}(\lambda=2.5)  & 85.30 & 53.80 & 49.66 \\\\\\hline
>   \text{ARoW}(\lambda=3.0)  & 84.65 & 54.23 & 50.11 \\\\\\hline
>   \text{ARoW}(\lambda=3.5)  & 83.86 & 54.13 & 50.15 \\\\\\hline
>   \text{ARoW}(\lambda=4.0)  & 83.73 & 54.20 & 50.55 \\\\\\hline
>   \text{ARoW}(\lambda=4.5)  & 82.97 & 54.69 & 50.83 \\\\\\hline
>   \text{ARoW}(\lambda=5.0)  & 82.53 & 55.08 & 51.33 \\\\\\hline
> \\end{array}
>
> - Also, in Figure 1 of the main manuscript, we can see that robust accuracies of ARoW is much higher than those of the competitors  under the same standard accuracy.
>
> 2-2. Would the proposed method improve the state-of-the-art model from [A]?
>
> [A] Uncovering the Limits of Adversarial Training against Norm-Bounded Adversarial Examples, 2021.
>
> -  In Table 3 of the main manuscript, we find that ARoW consistently outperforms [Rebuffi et al.] on various DNN architectures with extra data.
> - Since the method of [Rebuffi et al.] and [A] are almost identical except for a few experimental settings such as batch size, learning rate scheduler and regenerated extra data.
> - For this reason, we conjecture that comparison of ARoW with [A] is similar to the comparison with [Rebuffi et al.].

---

> > ### Comment · Reviewer_hmqr · 2022-11-16
> > **Thank you**
> >
> > I would like to thank the authors for their response and the additional experimental evaluations.
> > Thus, I will keep my initial positive score favoring the acceptance of this paper.

---

### Official Review · Reviewer_E21G · 2022-11-01

**Confidence:** 4
**Correctness:** 4
**Technical Novelty And Significance:** 3
**Empirical Novelty And Significance:** 3
**Recommendation:** 6

**Clarity, Quality, Novelty And Reproducibility:**

Clarity:

- The paper is clearly written and easy to follow.

Quality:

- The quality is good. The claims are clear and experiments are solid.

Novelty:

- Somewhat novel though similar to some previous methods.

Reproducibility:

- Code and implementation details are provided, so the reproducibility should be good. Though, I did not run the code to test.

**Strength And Weaknesses:**

Strength:

- Adversarial robustness is an important security issue in the field of deep learning. The proposed method pushes the SOTA performance of adversarial training to a new level.
- The proposed loss function is derived with a theoretical support.
- Extensive experiments show the empirical advantages of the proposed method from different aspects.

Weakness:

- The novelty of the method is ok, but it is similar to previous methods like MART. The author does point out the differences between the proposed method and other methods, so should not be a big problem.
- It's good that for table 1 and table 2 the results are based on 3 runs with standard errors given, but for most results, the improvements seem marginal.
- CIFAR10, F-MNIST, SVHN are all relatively small datasets. Does the method also perform well on larger dataset?


**Summary Of The Paper:**

This paper a new adversarial training method to improve the robustness of deep learning classifiers in the field of computer vision. To do so, the authors derived a new loss function, which is the surrogate of an upper bound of the robust risk. Specifically, they point out the differences and connections between the proposed method and previous adversarial training method. Experiments on three datasets (CIFAR10, F-MINIST, SVHN) show that the proposed method outperforms other baselines in terms of clean classification accuracy and robust accuracy. Ablation studies are done to show the effect of different parts of the loss function. They also show that the proposed method can be combined with other adversarial training techniques, such as extra data, to further improve the performance. Finally, experiments on CIFAR10 show that the proposed method is helpful to improve the fairness of the classifier compared to TRADES (which is an important baseline).

**Summary Of The Review:**

Overall it's a good paper with with theoretical justification and experimental support. Though, there are some weakness in terms of novelty and the limitation of the datasets used, it is a paper above the margin.

---

> ### Author Response · Authors · 2022-11-12
> **Response to reviewer E21G**
>
> We highly appreciate the reviewer E21G for the careful review and positive feedback.
>
> $\bf{\text{Weakness}}$
>
> 1. he novelty of the method is ok, but it is similar to previous methods like MART. The author does point out the differences between the proposed method and other methods, so should not be a big problem.
> - Yes, ARoW looks similar to MART, but they behave quite differently. In particular, ARoW dominates MART by large margins. The superiority of ARoW would be partly because ARoW is well theoretically motivated while MART is not.
>
> 2.  It's good that for table 1 and table 2 the results are based on 3 runs with standard errors given, but for most results, the improvements seem marginal.
> - For CIFAR10 with ResNet-18 in Table 1, ARoW outperforms HAT by 2.25\% points in standard accuracy while having +1.19\% improvement on PGD20 and similar robust accuracy for autoattack.
> - Around 2.25\% improvement can be favorably comparable to the 2.2\% improvement of HAT  over TRADES in standard accuracy while maintaining similar robust accuracies on autoattack reported in [1].
> - Moreover, this superiority is consistent with respect to DNN architectures and  benchmark datasets, and we can improve further by combining other methods such as AWP and FAT (Table 6).
> - For these reasons, we think that the empirical improvements of ARoW are significant.
>
> [1] Reducing Excessive Margin to Achieve a Better Accuracy vs. Robustness Trade-off, In ICLR, 2022.
>
> 3. CIFAR10, F-MNIST, SVHN are all relatively small datasets. Does the method also perform well on larger dataset?
> - For more complicated data, we analyzed CIFAR-100 in WRN-34-10 and found that ARoW still outperforms the other competitors.  See Table below. We will add this table and experimental setting in the revised version.
>
> \\begin{array}{|c|c|c|}
> \\hline  & \text{Stand} & \text{PGD}^{20} & \text{AutoAttack} \\\\\\hline
>   \text{PGD-AT}  & 62.20 & 32.27 & 28.66  \\\\\\hline
>   \text{TRADES} & 62.23 & 33.45 & 29.07  \\\\\\hline
>   \text{MART}  & 59.76 & 33.37 & 29.68 \\\\\\hline
>   \text{HAT}  & 60.42 & 33.75 & 29.42 \\\\\\hline
>   \text{ARoW}  & 62.83 & 34.74 & 30.42 \\\\\\hline
> \\end{array}
>
> - Note that most researches for adversarial learning do experiments with CIFAR10, MNIST, SVHN and CIFAR-100. An exception is Tiny-ImageNet which is 100,000 with 200 classes. Unfortunately, We could not analyze Tiny-ImageNet  because adversarial learning algorithms  require heavy computation and so the experiments could not be done within the rebuttal period.

---

### Decision · Program_Chairs · 2023-01-20

**Decision:**

Reject

**Justification For Why Not Higher Score:**

Limited novelty and the empirical results are not strong enough.

**Justification For Why Not Lower Score:**

N/A

**Metareview: Summary, Strengths And Weaknesses:**

The paper proposes a weighting mechanism to improve adversarial training. Although this is an interesting direction and there are several strengths pointed out by the reviewers (e.g., clear motivation, good related work section), during the discussion we identified several crucial weaknesses and thus decided to reject the paper:

- The novelty is limited:

The idea of "putting more regularization on less robust samples" has already been proposed in MART. As pointed out in the paper, the proposed formulation is almost identical to MART, except for the loss function and a small change to the weight term. However, it is not clear why these changes lead to improved performance. The authors only mentioned that their formulation is "theoretically motivated" which is not enough to explain the gain on top of MART. We suggest the authors carefully discuss these subtle differences and give intuitive explanations to clarify why the changes are important.

Further, Theorem 1 is also not very surprising since many previous papers like TRADES have already done similar decomposition.

- Empirical results are not strong enough:

As pointed out by some of the reviewers, the proposed method only marginally improves over existing ones, and the improvements become less clear when using more unsupervised data. This suggests that the improvements may not exist when combined with state-of-the-art methods (e.g., unsupervised data, augmentation, ...).

**Summary Of Ac-Reviewer Meeting:**

Reviewers actively participated during the AC-reviewer meeting --- they pointed out many strengths and weaknesses of the paper. Most of the reviewers think this is a borderline case given the limited novelty and marginal improvements in the experiments. The AC (who is an expert in machine learning robustness) also carefully went through the paper and thinks the algorithmic and theoretical contributions of this paper are quite limited, and the empirical results are not good enough to make the case for acceptance.